# Capturing a rhodopsin receptor signalling cascade across a native membrane

Siyun Chen[1,2], Tamar Getter[3], David Salom[3], Di Wu[1,2], Daniel Quetschlich[1,2], Dror S. Chorev[1✉], Krzysztof Palczewski[3,4,5,6✉] & Carol V. Robinson[1,2✉]

G protein-coupled receptors (GPCRs) are cell-surface receptors that respond to various stimuli to induce signalling pathways across cell membranes. Recent progress has yielded atomic structures of key intermediates[1,2] and roles for lipids in signalling[3,4]. However, capturing signalling events of a wild-type receptor in real time, across a native membrane to its downstream effectors, has remained elusive. Here we probe the archetypal class A GPCR, rhodopsin, directly from fragments of native disc membranes using mass spectrometry. We monitor real-time photoconversion of dark-adapted rhodopsin to opsin, delineating retinal isomerization and hydrolysis steps, and further showing that the reaction is significantly slower in its native membrane than in detergent micelles. Considering the lipids ejected with rhodopsin, we demonstrate that opsin can be regenerated in membranes through photoisomerized retinal–lipid conjugates, and we provide evidence for increased association of rhodopsin with unsaturated long-chain phosphatidylcholine during signalling. Capturing the secondary steps of the signalling cascade, we monitor light activation of transducin ($G_t$) through loss of GDP to generate an intermediate apo-trimeric G protein, and observe $G\alpha_t$•GTP subunits interacting with PDE6 to hydrolyse cyclic GMP. We also show how rhodopsin-targeting compounds either stimulate or dampen signalling through rhodopsin–opsin and transducin signalling pathways. Our results not only reveal the effect of native lipids on rhodopsin signalling and regeneration but also enable us to propose a paradigm for GPCR drug discovery in native membrane environments.

Molecular details of GPCRs, their G protein coupling and arrestin interactions are providing unprecedented insight into signalling cascades and are often achieved through judicious antibody stabilization or protein engineering[1]. Meanwhile, recognition of the importance of lipids in mediating GPCR signalling interactions has been derived from the use of lipid nanodiscs and peptidiscs to recreate membrane environments[5–7]. Capturing signalling of unmodified GPCRs, in native membrane environments, has so far eluded biophysical measurement. Here, selecting the best-characterized GPCR rhodopsin, the dim light receptor of the mammalian visual system[8], we formed lipid vesicles from disc membranes of dark-adapted bovine retinal rod outer segments (ROSs) and ejected this GPCR directly from fragments of its native membrane environment into a mass spectrometer (Fig. 1a–c, Supplementary Video 1). We monitored signalling by operating our mass spectrometer in a dark room and controlling light exposure to prompt photoisomerization of 11-*cis*-retinylidene (*cis*-retinal rho) to form activated all-*trans*-retinylidene (rho*). The Schiff base of rho* was then hydrolysed to the apo-protein opsin and all-*trans*-retinal[9,10]. During photoactivation, rho* interacted with GDP-bound transducin (heterotrimeric G protein ($G_t$)), leading to GDP–GTP exchange and

subsequent dissociation and formation of $G\alpha_t$•GTP, which then interacted with phosphodiesterase 6 (PDE6), displacing the PDE6 γ-subunit, releasing inhibition of catalysis and resulting in rapid hydrolysis of cytoplasmic cGMP[11]. Here we show that all components of this signalling pathway (protein, downstream effectors, cofactors and lipids) can be ejected simultaneously and their response to photon activation in the native membrane captured by native mass spectrometry directly and in real time (Fig. 1d).

To establish mass spectrometry and light conditions that enable monitoring of rhodopsin signalling, and before instituting our approach in native membranes, we used membrane solubilization and detergents to extract rhodopsin from dark-adapted ROS membrane, as previously described for analysis via mass spectrometry[12–14]. Following extraction into lauryl maltose neopentyl glycol (LMNG), we identified illumination conditions (LED-emitting cold white light; Extended Data Fig. 1) and mass spectrometry conditions to release rho from membrane lipids and detergents and to monitor rho* signalling through the mass change associated with conversion of rho to opsin (Δ mass = 266 Da). Before illumination, 73 ± 1% rho (11-*cis*-retinal or *trans*-retinal) and 27 ± 1% opsin were present. Recording of spectra in real time was synchronized with

[1]Chemistry Research Laboratory, University of Oxford, Oxford, UK. [2]Kavli Institute for Nanoscience Discovery, University of Oxford, Oxford, UK. [3]Gavin Herbert Eye Institute, Department of Ophthalmology, University of California, Irvine, Irvine, CA, USA. [4]Department of Physiology and Biophysics, University of California, Irvine, Irvine, CA, USA. [5]Department of Chemistry, University of California, Irvine, Irvine, CA, USA. [6]Department of Molecular Biology and Biochemistry, University of California, Irvine, Irvine, CA, USA. ✉e-mail: drorchorev@gmail.com; kpalczew@uci.edu; carol.robinson@chem.ox.ac.uk

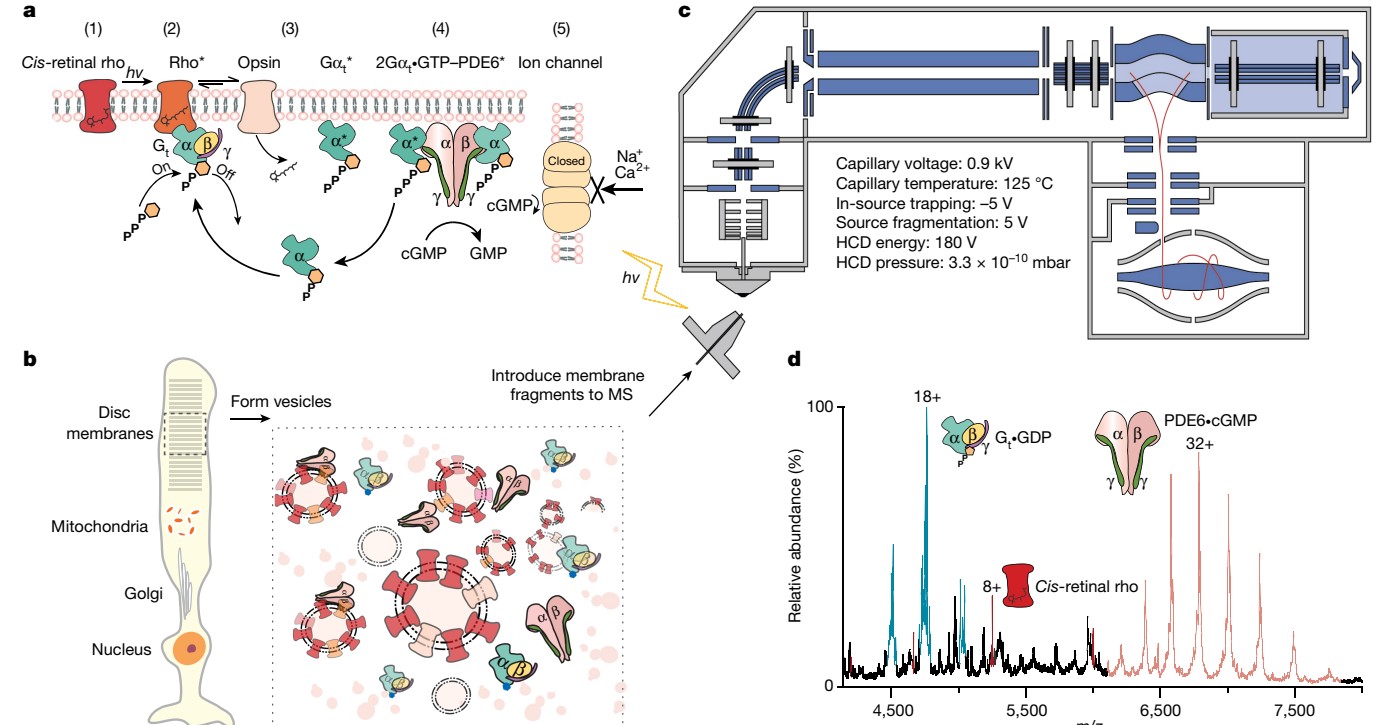

**Fig. 1 | Established signalling pathway of rho, experimental conditions for preparation of vesicles for GPCR signalling and representative mass spectrum. a**, Following absorption of a photon of light *hν* (1), 11-*cis*-retinal of rho isomerizes to the all-*trans* isomer (2). The activated states of rho (rho*) engage with transducin (G_t), consisting of Gα_t•GDPβγ_t, and exchange GDP for GTP. G_t dissociates to form Gα_t•GTP and Gβγ_t; loss of retinal from rho* leads to the formation of opsin (3). α-Subunits of G_t interact with the γ-subunits in the PDE6 enzyme, with γ-subunits undergoing a conformational change, relieving inhibition and thereby activating PDE6 to cause hydrolysis of cGMP (4). Depletion of cGMP then closes the ion channel and the 'dark current' is terminated (5). The resulting change in the membrane potential produces the sensation of light. **b**, Disc membranes of rod cells are homogenized to form a

heterogeneous distribution of vesicles that are introduced directly into the mass spectrometer (MS). The spectrum shown was recorded using the parameters stated above, which led to the dissociation of lipids from proteins. **c**, An LED light source is configured to apply timed light intervals before the electrospray ionization of the vesicles into an Orbitrap mass spectrometer under the conditions noted. HCD, higher-energy collisional dissociation. **d**, Following the addition of a soluble fraction containing PDE6 and G_t, all proteins along the signalling pathway were detected. The proteins were ejected intact as rho/opsin (red), trimeric G_t•GDP (cyan) and tetrameric PDE6•cGMP (pink). The mass spectrum shown from *m/z* 4,000 to *m/z* 8,000 represents the raw data. The experiment was repeated three times.

illumination; equal population of both species was achieved after 3 min (Fig. 2a), reaching a plateau with $27 \pm 1\%$ rho remaining at 20 min in this detergent micelle preparation (Fig. 2c). We next established mass spectrometry conditions that enabled ejection of rho and opsin (rho/opsin) directly from disc membrane fragments while activated with light. To do this, we adapted and applied a sonicated lipid vesicle approach[15,16] to purified ROS disc membranes, forming vesicles for direct introduction into the mass spectrometry. Under these conditions, monomeric rho was ejected. No dimeric population was observed, attributed to the relatively small dimer interface observed via cryo-electron microscopy in nanodiscs[17] that dissociates under these mass spectrometry conditions. Comparing the post-translational modification status of dark-adapted and light-exposed rho/opsin extracted in detergent or ejected from membranes, we found that glycosylation, palmitoylation and cysteinylation status are largely indistinguishable (Extended Data Fig. 2a–d, Extended Data Table 1). An increase in phosphorylation of opsin was observed 20 min after signalling was initiated, both in detergent-extracted and membrane-ejected rho/opsin, consistent with progress towards termination of this signalling pathway through eventual interaction with arrestin[18].

Monitoring rho/opsin populations during continuous light exposure in membrane vesicle fragments, we found that after 3 min rho remained predominant over opsin (Fig. 2b). The reaction in membranes proceeds until a plateau is reached with approximately 32% rho remaining after 20 min (compared with approximately 27% in detergent micelles

after the same time period (Fig. 2c)). We next calculated and compared the rates of chromophore hydrolysis ($k_{hyd}$; at pH 7.0 and 28 °C) in LMNG micelles and from ROS membrane vesicles according to a kinetic model (Fig. 2c, Extended Data Fig. 3, eq. 7). As the hydrolysis of all-*trans*-retinal from rho* is a slower step than photoisomerization of 11-*cis*-retinal to *trans*-retinal in rho, our results allow us to conclude that hydrolysis of all-*trans*-retinylidene from rho* is slower in vesicles than in detergent micelles. Considering the isomerization reaction, loss of all-*trans*-retinal from the binding pocket of rho* is monitored through a change in mass following hydrolysis of the Schiff base. The extent of *cis*-retinal or *trans*-retinal bound to opsin before hydrolysis, however, is not apparent via mass alone. To delineate these isomeric populations, we adopted a well-established protocol in which ROS membrane preparations were pre-incubated in the dark with 5 mM hydroxylamine (Fig. 2d). Rho in its ground state is insensitive to hydroxylamine but, upon photoisomerization of *cis*-retinal to all-*trans*-retinal, hydroxylamine is able to attack and cleave the Schiff base in the photoactivated rhodopsin to form all-*trans*-retinyl oxime[19]. As expected, rho* in membranes decayed significantly faster in the presence of hydroxylamine (Extended Data Fig. 2e). This hydroxylamine experiment, together with the rho/opsin photoconversion of untreated membranes mentioned above ($k_{hyd}$), enabled us to subtract the two decay curves and consequently obtain the rates of retinal isomerization ($k_{iso}$) and hydrolysis without isomerization ($k'_{hyd}$) in native membranes (Fig. 2e, Extended Data Fig. 3).

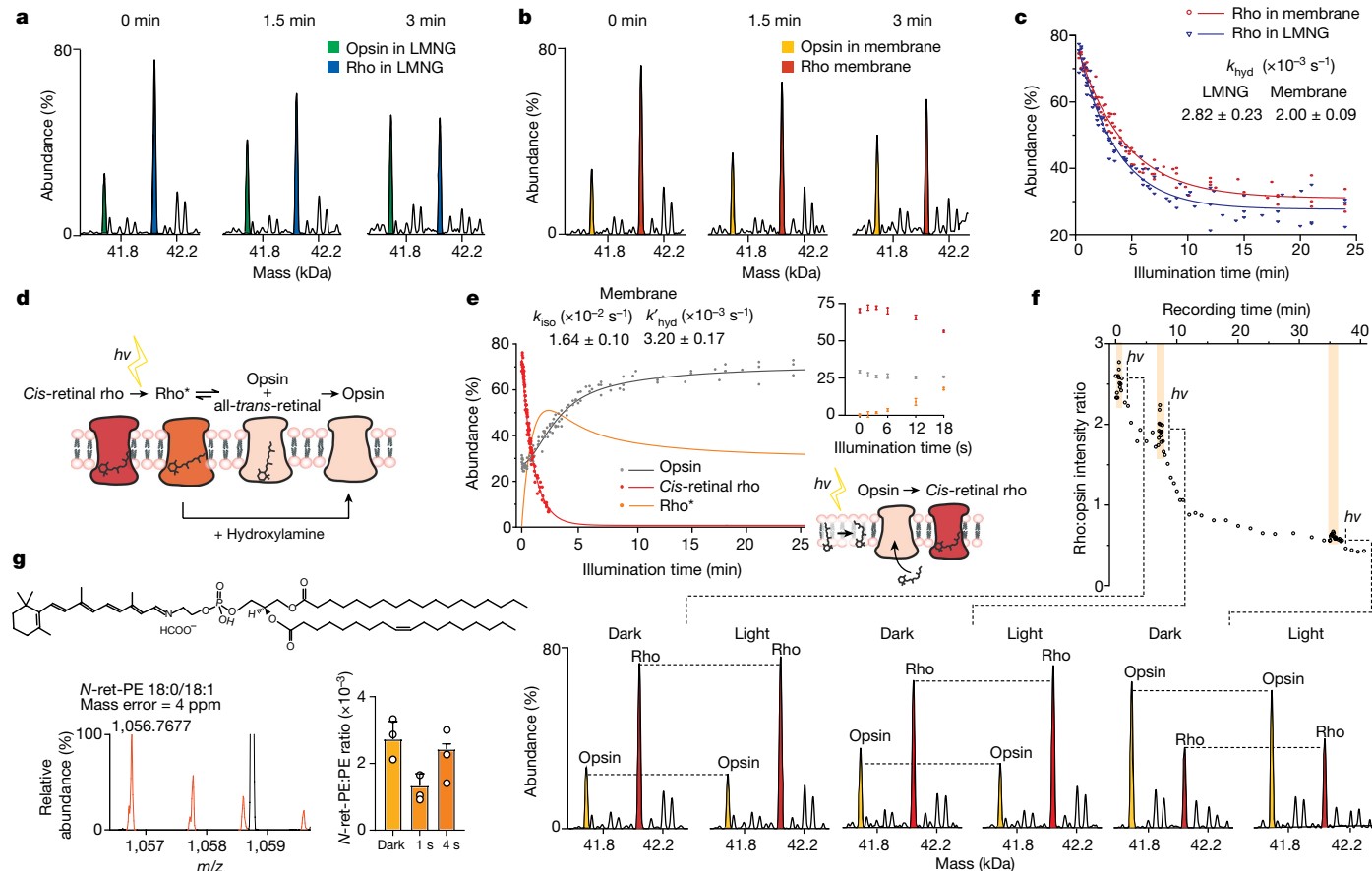

**Fig. 2 | Monitoring real-time conversion of rho to opsin in detergent micelles and in native membranes, probing the effects of pulses of light and confirming the formation of *N*-ret-PE. a**, **b**, Changes in the population of rho and opsin in LMNG (blue and green, respectively; **a**) and in native membranes (red and yellow, respectively; **b**). Individual spectra are shown as zero-charge plots with illumination times stated. **c**, Plot of the relative abundance of rho during illumination, in either LMNG detergent micelles (blue) or native membranes (red) monitored as a change in mass as a function of time. Rate constants for hydrolysis are shown for the reaction in detergent micelles or in membranes (Extended Data Fig. 3). **d**, Schematic of the dark-adapted state undergoing light-activated conversion of *cis*-retinal to all-*trans*-retinal, hydrolysis of the Schiff base and dissociation from rho to form opsin. **e**, Monitoring the decay of *cis*-retinal rho in membranes pretreated with

hydroxylamine (red), the generation of opsin in native membranes (grey) and the change of rho* calculated over the illumination period (orange). The inset, which is an expansion of the illumination period from 0–18 s, shows a less than 18-s increase in rho abundance at the expense of opsins. The schematic depicts a possible regeneration mechanism through isomerization of *trans*-retinal. **f**, Monitoring the conversion of rho to opsin and the regenerative reaction of opsin to rho following pulses of light at 0.51 min, 7 min and 35 min for time periods of 3 s, 12 s and 32 s, respectively. Zero-charge mass spectra are shown at time points during this conversion reaction, before illumination (dark) and at the peak of the three light pulses. **g**, Spectrum (orange) of *N*-ret-PE (18:0/18:1 representative structure shown) extracted from native membranes; the bar graph shows changes in the equilibrium between conjugated and free PE in response to illumination time. Data are presented as mean ± s.d. (*n* = 3).

Careful examination of the data reveals an initial phase in which opsin regenerates to rho (less than 18 s) in competition with the overall rho/opsin conversion. The increase in *cis*-retinal rho at 2.7 ± 1 s is coincident with the lowest level of opsin at 3.6 ± 1 s (Fig. 2e, inset). This initial reaction, in the absence of hydroxylamine, must therefore arise primarily from *cis*-retinal binding to opsin. To determine whether this observation was a feature of the membrane, we applied short-term illumination on disc membrane preparation and to rho purified in LMNG detergent micelles. Upon each transient illumination of rho from ROS, a rapid rise of rho was observed together with the general trend of rho transforming to opsin (Fig. 2f). Some evidence for regeneration of rho in micelles was observed, although to a much lower extent (Extended Data Fig. 4). The ability to regenerate is a known property of the visual system, which requires the conversion of all-*trans*-retinal to 11-*cis*-retinal and has been shown to be carried out by enzyme pathways in neighbouring cells[10]. Our proteomics experiments (Supplementary Table 1) were unable to detect key enzymes from the regeneration cycle, such as retinal GPCR (RGR) or the retinoid isomerase RPE65 (ref. [20]), prompting us to consider the possibility that this rapid regeneration (less than 18 s) may be a feature of the membrane itself.

Previous reports have implicated an *N*-all-*trans*-retinyl-PE conjugate that can undergo photoisomerization to form *cis*-retinals (primarily 11-*cis*-retinal (85–86%) with small contributions from 9-*cis*-retinal (12–13%) and 13-*cis*-retinal (1–2%))[21]. Although our mass spectrometry approach cannot readily distinguish these structural isomers, our lipidomics data reveal the presence of high concentrations of polyunsaturated fatty acids including PE (Extended Data Fig. 5a–c). To explore the formation of retinal conjugated to PE, we extracted lipids into isopropanol directly from dark-exposed and controlled-light-exposed ROS disc membranes. We selected three representative lipids (PE 18:0/18:1, 18:0/20:4 and 18:0/22:5) with minimal overlap in mass spectra, and searched for their respective retinyl conjugation. We found direct evidence for the formation of *N*-retinyl-PE (*N*-ret-PE) in all cases and monitored its change as a function of exposure to light. After 1 s of illumination, a significant decrease in conjugation was observed (more than 2.5-fold for PE 18:0/18:1), consistent with light-stimulated retinal dissociation from PE (Fig. 2g, Extended Data Fig. 5d–f). Peaks assigned to *N*-ret-PE increased after 4 s of light exposure to similar levels as the dark-adapted membrane, suggesting further conjugation events.

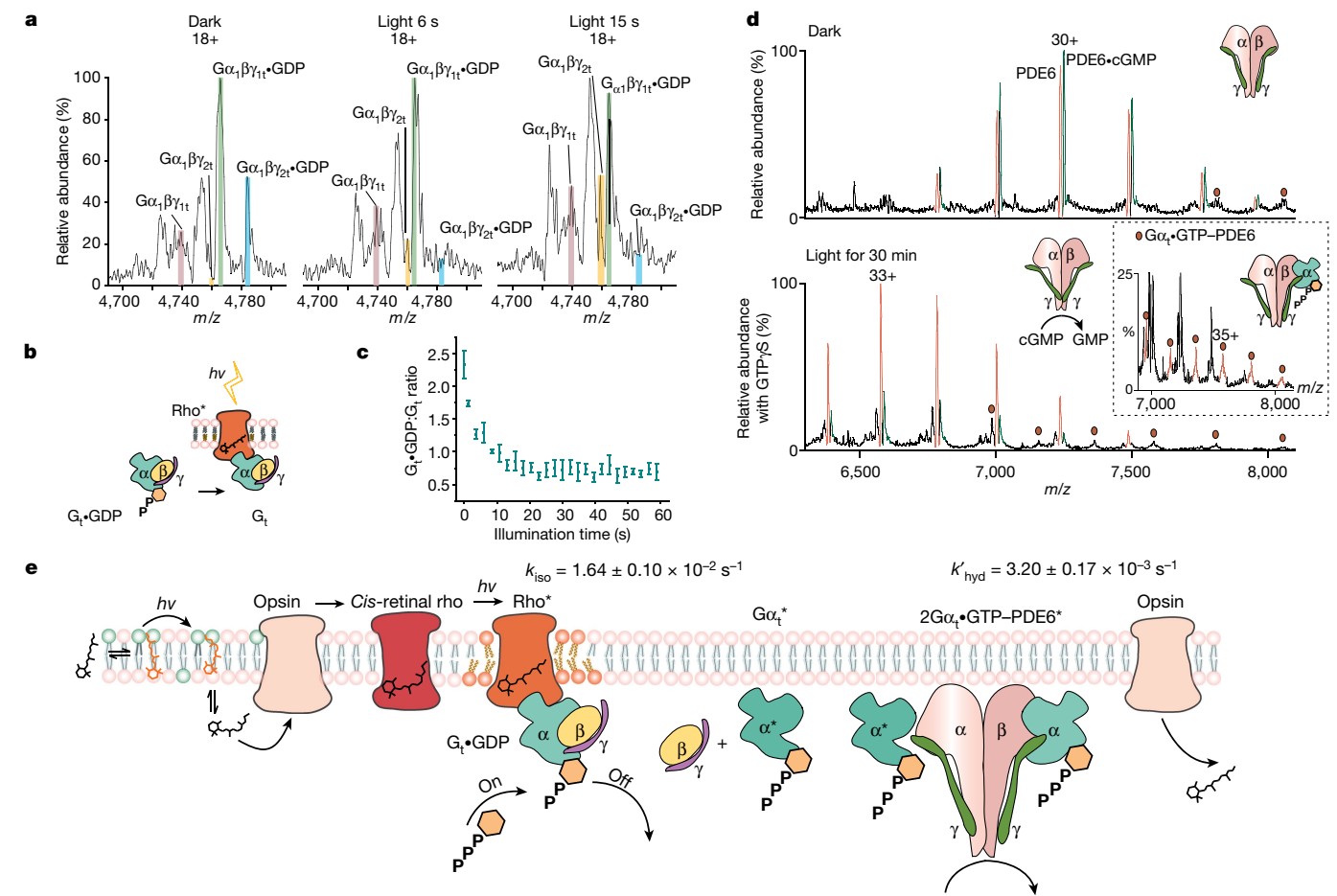

**Fig. 3 | Light activation of rho in ROS disc membrane signalling through $G_t$ to release $G\alpha_t \cdot GDP$ to effect hydrolysis of cGMP and its release from PDE6.** **a**, Native mass spectra of $G_t \cdot GDP$ with no additional GTP added containing two α-isoforms ($\alpha_1$ and $\alpha_2$) and two γ-isoforms ($\gamma_1$ and $\gamma_2$) are catalysed to apo-$G_t$ during a 15-s time course of illumination. **b**, Reaction scheme: rho* catalysed the nucleotide exchange of $G_t$. **c**, The ratio of ground-state $G_t \cdot GDP$ to intermediate apo-$G_t$ decreases rapidly during the first 15 s after illumination. Data are presented as mean ± s.e. ($n = 3$). **d**, Native mass spectra of PDE6 ejected from ROS disc membrane vesicles under dark conditions reveal that cGMP binds to intact tetrameric PDE6 with an approximately 1:1 ratio (top). A low population of PDE6–$G\alpha_t \cdot GTP$ is also observed (red circles). After exposure to light in the presence of ROS disc membranes, the population of PDE6 bound to cGMP is reduced considerably, consistent with the release of GMP following its

hydrolysis via activated PDE6 with addition of a molar equivalent of GTPγS to $G_t$ (bottom). **e**, Schematic shows the rho* signalling cascade that involves the conjugation of all-*trans*-retinal with PE and its light activation to form *cis*-retinal that is able to interact with opsin. Light-activated conversion of rho to an intermediate state takes place with rho* capable of interacting with $G_t$. Changes in the lipid bilayer are depicted as unsaturated lipids that are recruited during rho* signalling (lipids, orange). The $G\alpha_t \cdot GTP$ subunit produced following hydrolysis of $G\alpha_t \cdot GDP$ interacts with PDE6, relieving its inhibition via the γ-subunit and effecting the hydrolysis of cGMP, which is then released from PDE6. $G\alpha_t \cdot GDP$ is formed for the regeneration of $G_t \cdot GDP$. All experiments shown in this figure were repeated at least three times. Data are presented as mean ± s.e. ($n = 3$).

Significantly reduced rho regeneration was observed for rho purified in micelles, where a lack of the conjugation system for retinal and PE would be anticipated (Extended Data Fig. 4). Moreover, regeneration of rho might be expected to be limited by the availability of *cis*-retinal. To investigate this possibility, we incubated intact ROS disc membranes with a tenfold excess of all-*trans*-retinal. Following illumination, regeneration of rho from 79% to 91% was observed after 3.8 min (Extended Data Fig. 6). Together, these results imply that conversion of opsin to rho in membranes can be supplemented by the addition and photoisomerization of all-*trans*-retinal and that conjugation of PE provides a source of *cis*-retinal for regeneration in membranes, a mechanism that has been suggested for sustained vision in daylight[21].

Turning our attention to the next step in the signalling cascade, rho* signals through $G_t$, releasing GDP and dissociating to form $G\alpha_t \cdot GTP$ for interaction with PDE6. Before illumination, GDP-bound heterotrimeric $G_t \cdot GDP$ and cGMP-bound heterotetrameric PDE6 (PDE6·cGMP) were observed as the predominant species (Fig. 1d). To monitor real-time

signalling through transducin, excess soluble fraction containing $G_t$ and PDE6 was added to our membrane preparation (see Methods). We monitored the decrease in the ratio of $G_t \cdot GDP:G_t$ as a function of time, initially for periods of up to 1 min (Fig. 3c), but found little change beyond the first 15 s of light stimulation. The spectra of $G_t$ are complicated by the presence of additional proteoforms ($\alpha_2$ from cone $G_t$ (Extended Data Fig. 7a–c)). Control experiments in the absence of light or ROS membranes (Extended Data Fig. 7d–f) confirm that light activation of rho in the membrane is required to decrease the population of $G_t \cdot GDP$ during signalling (Fig. 3a).

To explore the relationship between Gt-dependent rho* signalling and PDE6·cGMP hydrolysis, we investigated three different GTP conditions: (1) with endogenous levels of GTP in the soluble fraction, (2) following the addition of a molar equivalent of GTP to $G_t$, and (3) in the presence of a molar equivalent of non-hydrolysable GTPγS to $G_t$. After illumination, the levels of $G_t \cdot GDP$ decreased in all three cases but were replenished in condition 2, the supplementary GTP experiment

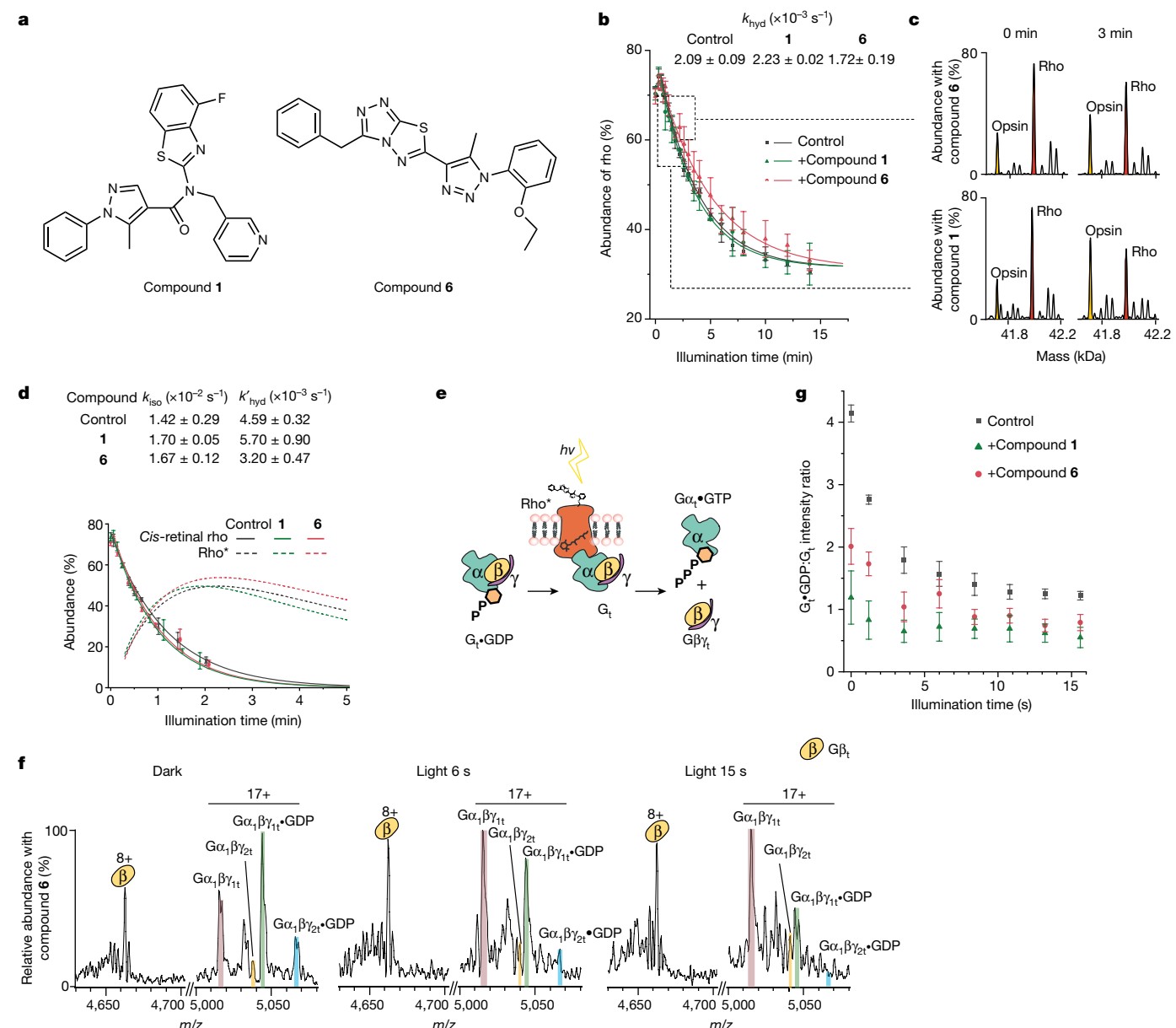

**Fig. 4 | The effects of rho-targeting compounds on the rho to opsin conversion and the hydrolysis of $G_t \cdot GDP$. a**, Structures of the two rho-targeting compounds (**1** and **6**) used here. **b**, Monitoring the conversion of rho to opsin via the change in mass following illumination in the presence of **1** or **6** and in a control under the same conditions. Data are presented as mean ± s.e. ($n = 3$). **c**, Time points for spectra of rho (red) and opsin (yellow) ejected from membranes at 0 and 3 min in the presence of **1** (bottom) and **6** (top) are shown on a zero-charge scale. **d**, Abundance of *cis*-retinal rho and rho* as a function of time in the absence (grey) or presence of **6** (pink) or **1** (green). The function for *cis*-retinal rho is derived from fitting data from more than 3.6 s (dashed lines)

and the function for rho* is derived from the fit of the data from 18 s to 300 s (solid lines) and extrapolated (dotted lines). Data are presented as mean ± s.e. ($n = 3$). **e**, Reaction scheme: following activation with light, rho* signals through $G_t$ to the nucleotide-free form ($G_t$), which then dissociates to form βγ and α. **f**, Mass spectra of $G_t$ in the presence of compound **6** added to the native membrane in the absence of light (dark), and after exposure to light for 6 s and 15 s. **g**, Monitoring signalling through $G_t$ via changes in the ratio of $G_t \cdot GDP:G_t$ in the presence or absence of compound **1** or compound **6** as a function of illumination time. Data are presented as mean ± s.e. ($n = 3$).

(Extended Data Fig. 7g–i). Linking $G_t$ signalling with PDE6•cGMP hydrolysis, we monitored the release of the hydrolysed GMP product via the ratio PDE6:PDE6•cGMP. For dark-adapted membranes, before illumination, a PDE6:PDE6•cGMP ratio of approximately 1:1 was observed, consistent with full occupancy of one substrate-binding site in PDE6 (Fig. 3d, top). Signalling of rho* through $G_t$ prompts further hydrolysis of cGMP with supplementary GTP (PDE6:PDE6•cGMP ratio of approximately 1:0.25) compared with endogenous levels (PDE6:PDE6•cGMP ratio of approximately 1:0.85) (Extended Data Fig. 7g, h). As $G\alpha_t \cdot GTP\gamma S$ is also able to interact with PDE6 (ref. [22]), an intermediate level of

hydrolysis of cGMP was observed in the presence of an equimolar aliquot of $G\alpha_t \cdot GTP\gamma S$ (PDE6:PDE6•cGMP of approximately 1:0.4) (Fig. 3d, bottom, Extended Data Fig. 7i). An additional PDE6–$G\alpha_t \cdot GTP$ complex with 1:1 stoichiometry can also be discerned at low intensity in the presence of $G\alpha_t \cdot GTP\gamma S$ (Fig. 3d, inset). Under these experimental conditions, 1:2 complexes of PDE6•($G\alpha_t \cdot GTP)_2$ and PDE6•(cGMP)$_2$ were not observed, consistent with existing mechanistic models. According to one model, although controversial[23], one catalytic subunit of PDE6 binds to $G\alpha_t \cdot GTP$ with high affinity but low activity for hydrolysing cGMP, whereas the other catalytic subunit of PDE6 binds to $G\alpha_t \cdot GTP$

with low affinity but is able to hydrolyse cGMP at a high rate[11]. Our data, with only one Gα[t]•GTP subunit binding to PDE6 and with a high intensity for a single binding site of cGMP, are in accordance with the asymmetry of the activated complex and its ability to hydrolyse one cGMP to GMP per PDE6 tetramer. This reaction in turn is regulated by available GTP, activation of rho* through G[t] and causes decreased concentrations of cGMP for closure of the ion channel.

Having established a means of capturing the entire signalling cascade we considered further the influence of the membrane through the effect of lipid microenvironments on rho* signalling through G[t]. We examined lipids in the spectrum of dark-adapted rho expelled from membranes in the presence of G[t] and PDE6 (see Methods) and found a predominance of long-chain polyunsaturated phosphatidylcholinelipids (PC 40:6, 40:8, 38:6, 38:8, 34:1, 34:2) (Extended Data Fig. 8a) in line with observations of distinct regions of retinal rods in nanodisc experiments[7]. After exposure to light (6–18 s) in the presence of G[t] and PDE6, we observed a notable increase in the ratio of unsaturated to saturated PC lipids compared with initial and later time points (Extended Data Fig. 8a, b). Initial contact with unsaturated lipids is in line with findings of a molecular dynamic simulation study of a GPCR in an inactive or partially active state receptor[24] and NMR experiments[25]. The increase of unsaturated lipid chains associated to rho during light activation provides evidence to support earlier proposals[24] that polyunsaturated chains associate specifically with rho* to enable its conformational change from its ground state and thereby facilitate swift signalling.

Monitoring signalling in a membrane environment provides us with a unique opportunity to probe the effects of rho-targeting molecules, identified through cell-based assays designed to select compounds that enhance or perturb rho dimerization[26,27] (see Methods). We tested nine compounds individually and measured their effect on rho $k_{hyd}$ against a control (compounds 1–9; Extended Data Fig. 9a, b). The results allowed these ligands to be divided into two groups that were capable of either accelerating with a marginal effect or, more commonly, slowing retinal hydrolysis. Compounds 1 and 6 represent these two capabilities and were selected for further study (Fig. 4a). Clear differences emerge post-illumination (3 min) as rho and opsin predominate in the presence of 1 and 6, respectively (Fig. 4b, c). After 15 min, noticeable differences occur in the rho to opsin conversion, which is faster when modulated by 1 than by 6, with 1 being marginally faster than the control. As we see no evidence for displacement of retinal by these compounds or changes in the conformation of rhodopsin[27], we speculate that 1 and 6 act as allosteric modulators. Calculating rates of hydrolysis and isomerization in the presence of hydroxylamine, we found that 1 accelerates both hydrolysis and isomerization, whereas rho* bound to 6 has an accelerated isomerization rate but retains retinal by decelerating hydrolysis, thereby potentially maintaining active signalling states for extended periods.

Anticipating that the prolonged activation state of rho* in the presence of 6, and the rapid isomerization and hydrolysis of 1, would affect signalling differently through G[t], we established a further assay in which we measured the ability of 1 and 6 to affect phototransduction. As above, we monitored their impact by measuring the change in mass of the conversion of G[t]•GDP to apo-G[t]. Before illumination, in the presence of 1 or 6 and a control under dark conditions, the ratio of G[t]•GDP:G[t] was 1.2, 2.0 and 4.1, respectively. In the presence of the ROS membrane, both ligands had caused significant conversion of G[t]•GDP:G[t] compared with the control (Fig. 4f, Extended Data Fig. 10). Unexpectedly, 1 and 6 induced transduction in the dark, implying that their enhanced isomerization rates enable them to signal through the initial *trans*-retinal rho population before illumination. Once illuminated, phototransduction proceeded further (G[t]•GDP:G[t] ratios of 0.6, 0.8 and 1.2 after 15 s for 1, 6 and the control, respectively; Fig. 4g). We conclude that 1 and 6 target rho and amplify signalling through G[t], making them ideal starting points for further chemical development. More generally, the fact that we can separate these reactions into isomerization, hydrolysis and

influence on a downstream effector enables us to study rho-targeting compounds in molecular detail via their effects on phototransduction, in real time and within their native membrane environments.

Together, our native mass spectrometry data document that we have conserved a signalling cascade for an archetypal class A GPCR across its native membrane. We have shown that all three key players in the cascade can be ejected simultaneously from vesicles formed from native ROS disc membranes. In so doing, we have captured the conversion of rho to opsin in real time and have shown that it is significantly slower in the membrane than in the detergent micelle. We observed regeneration of rho following exposure to light and detected changes in the level of *N*-ret-PE and depletion of the conjugated chromophore following illumination. The importance of the membrane also prompted us to consider the changing lipid microenvironment during signalling and documented changes in the association of unsaturated long-chain PC lipids in the vicinity of signalling rho. Considering the effects of rho-targeting compounds, we demonstrated their ability to accelerate the rho to opsin conversion or slow down the reaction, and we hypothesize that the latter would stabilize a signalling-competent state. In line with this hypothesis, we demonstrated the ability to modulate the G[t] signalling pathway through an increase in the turnover of G[t]•GDP to G[t] and the subsequent dissociation to the Gβγ[t] complex and Gα[t]•GTP primed for interaction with PDE6. We observed the static intermediate Gα[t]–PDE6, as well as the effects of endogenous GTP levels on the hydrolysis of cGMP through light-activated signalling across the membrane. By capturing the entire signalling process, we have demonstrated the importance of the lipid microenvironment for signalling, coupling, effector targeting and regeneration. We anticipate that similar approaches will be applicable to other cell-surface receptors (for example, olfactory receptors) present at high density, enabling kinetic studies of molecular events, and identification of endogenous and synthetic receptor ligands that perturb signalling. Overall, we have highlighted an approach to drug discovery in which different stages of a signalling cascade can be targeted across native membranes.

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

## Methods

### Preparation of bovine rod disc membrane vesicles, detergent solubilized rho and purified rho

Bovine eyes were obtained from a commercial slaughterhouse. ROSs were obtained from a batch of 50–100 eyes each time, with dark-adapted retinas, and purified as previously described[28,29]. To prepare vesicles, rho disc membranes containing 81 μg rho were suspended in 250 μl of 200 mM ammonium acetate. An earlier protocol was adapted[16] to homogenize disc membranes using a probe sonicator with a stepped tip microtip (2 mm; Vibra-Cell VCX-500 Watt, Sonics) and a maximal amplitude (40%) (1 s on, 2 s off) applying 2 J per cycle for 1.5 min. Disc membrane vesicles containing rho (approximately 9 μM) were ready for study via native mass spectrometry (MS) either directly, or alternatively following addition of hydroxylamine to a final concentration 5 mM, at pH 7.0.

For detergent-solublized rho, collected ROS was solubilized with 20 mM LMNG overnight. The solution was then diluted into 50 μl 200 mM ammonium acetate with 0.004% LMNG and buffer exchanged with 200 mM ammonium acetate and 0.002% LMNG using a micro bio-spin column (6,000 Da MW cut-off) before native MS. The resulting solution (approximately 9 μM rho in LMNG ammonium acetate solution) was exposed to light for MS measurement of rho/opsin kinetics (Fig. 2).

For rho purification, to remove the majority of lipids for the regeneration experiments (Extended Data Fig. 4), ROS, washed with isotonic and hypotonic buffers, was solubilized in 5 mM LMNG. Rho was then purified with an immobilized 1D4 antibody as previously described[30], with the modification that LMNG was used as the detergent. Purified rho was then dialysed (10 kDa cut-off) to remove the nonapeptide TETSQVAPA.

All above procedures were carried out in the dark under dim red light (more than 670 nm).

### Purification of bovine rod $G_t$ and PDE6

Trimeric $G_t$ and PDE6 were prepared as previously described[31]. In brief, bovine ROSs were suspended in isotonic buffer containing 20 mM 4-(2-hydroxyethyl)-1-piperazineethanesulfonic acid (HEPES), pH 7.5, 100 mM NaCl, 1 mM DTT and 5 mM MgCl$_2$ and centrifuged at 31,000$g$ for 25 min at 4 °C. The pellet was homogenized in a hypotonic buffer containing 5 mM HEPES, pH 7.5, 1 mM EDTA and 1 mM DTT and centrifuged at 40,000$g$ for 30 min at 4 °C multiple times. The ratio between $G_t$ and PDE6 in the supernatant was analysed by ImageJ software on a Coomassie blue staining SDS–PAGE. For addition of the soluble fraction to the membrane, $G_t$ and PDE6, as purified above, were first buffer exchanged three times with 200 mM ammonium acetate using an Amicon filter (10 kDa cut-off). Under dim red light, rho, $G_t$ and PDE6 were mixed at a molar ratio of 13:4:1 for 15 min on ice to allow diffusion of $G_t$ and PDE6 before analyses by native MS. To capture post-signalling PDE6 for native MS, rho, $G_t$ and PDE6 at a molar ratio of 32.5:4:1 were illuminated under cold white LED for 30 min at room temperature and centrifuged at 20,000$g$ for 20 min to remove the membrane (presented in Fig. 3d, Extended Data Fig. 7g–i).

### Native MS set-up and illumination conditions

The native membrane preparation (2–3 μl) was placed directly in a gold-coated nano-electrospray capillary before the source of the mass spectrometer (Fig. 1c). The cold LED light source (120 μW at a 10 cm distance with a surface area of illumination of 3 mm$^2$) was assembled via a bracket to illuminate the tip of the capillary for a defined interval in a dim red-light environment. For a real-time dynamic process, the membrane was first recorded in the dark and then illuminated with light pulses, or continuously, depending on the experiment. Native MS experiments were carried out on a Q-Exactive adapted for membrane proteins[32]. The following parameters were used typically and adjusted to release the membrane protein from the detergents and

membrane lipids: capillary voltage of 0.8–1.4 kV, capillary temperature of 125–200 °C, higher-energy collisional dissociation (HCD) energy of 125–200 V, desolvation voltage of −5 to 0 V, source fragmentation of 5–100 V, HCD pressure of $3.8 × 10^{-10}$ to $8.5 × 10^{-10}$ mbar, C-trap entrance lens tune offset was set to 2, injection flatapole was set to 7 V, inter-flatapole lens was at 0 V, and the bent flatapole was set at 1 V. Threshold was set to 3. These MS conditions were modified to focus on the individual components of the spectrum. For example, the following parameters were used for Fig. 1d: capillary voltage of 0.9 kV, desolvation voltage of −5 V, source fragmentation of 5 V and HCD energy of 180 V. HCD pressure was typically $3.3 × 10^{-10}$ mbar. The spectra of rho/opsin (Fig. 2) were collected at a capillary voltage of 1.1 kV, desolvation voltage of 0 V, source fragmentation of 100 V and HCD energy of 200 V. HCD pressure was typically $5.5 × 10^{-10}$ mbar. For optimized spectra of $G_t$ (Figs. 3c, 4f) the following parameters were used: capillary voltage of 0.9–1.1 kV, desolvation voltage of 0 V, source fragmentation of 25 V and HCD energy of 175 V. HCD pressure was typically $8.5 × 10^{-10}$ mbar. For the measurement of PDE6 spectra (Fig. 4d, e), the following parameters were used: capillary voltage of 1.1 kV, desolvation voltage of 0 V, source fragmentation of 50 V and HCD energy of 175 V. HCD pressure was typically $7.3 × 10^{-10}$ mbar.

### Polyunsaturated PC lipids surrounding rho identified with multistage native MS$^n$

The mixture of rho, $G_t$ and PDE6 at a molar ratio of 32.5:4:1 (described above) was used to identify the polyunsaturated PC surrounding rho during light-induced signalling. PC lipids are released from rho and detected in positive mode using an Orbitrap Eclipse tribrid mass spectrometer (Thermo Fisher Scientific)[33]. In general, in-source activation (100 V) was applied to rho in membranes and the resulting ions were transferred to the ion-routing multipole (IRM) for activation (HCD normalized collision energy (NCE) of 100%, IRM pressure of 8 mM) to promote dissociation into lipids. Automatic gain control (AGC) target values (100–150%) and maximum injection time (100 ms) were adjusted manually to maximize the normalized level. Detection at this stage was typically performed in the Orbitrap at a high $m/z$ range 500–8,000. High resolution at 60,000 can be used to enhance the intensity of lipid signals at low $m/z$. Data were analysed using the Xcalibur software package 4.1 (Thermo Fisher Scientific).

### Data analysis

Raw data were first analysed manually using Xcalibur 4.1 (Thermo Fisher Scientific). The relative abundance of each species in a real-time measurement was quantified using Lig2Apo, a simple jupyter notebook. A folder of text files (format $m/z$ versus intensity) was exported from Xcalibur 4.1 (Thermo Fischer Scientific) and two series of $m/z$ values were defined corresponding to the molecular species of interest. The program then read all text files and calculated the relative intensity of the series and the intensity ratio of two species (see Code availability). The zero charge spectra presented in Fig. 2 were analysed by Unidec v.2.7.3[34]. The rate constants were analysed by OriginPro 2020 SR1 9.7.0.188.

### Proteomics and protein identification

Protein bands were excised from gels and processed as previously described[35]. Peptides generated were resuspended in 0.1% formic acid and separated on an Ultimate 3000 UHPLC system (Thermo Fisher Scientific) and electrosprayed directly into a Q Exactive mass spectrometer (Thermo Fisher Scientific) through an EASY-Spray nano-electrospray ion source (Thermo Fisher Scientific). The peptides were trapped on a C18 PepMap100 pre-column (300 μm i.d. × 5 mm, 100 Å; Thermo Fisher Scientific) using solvent A (0.1% formic acid in water). The peptides were separated on an analytical column (75 μm i.d. packed with ReproSil-Pur 120 C18-AQ, 1.9 μm, 120 Å, by Dr Maisch GmbH) using a gradient (15–38% for 30 min, solvent B – 0.1% formic acid in acetonitrile, flow rate:

200 nl min$^{-1}$) for 15 min. The raw data were acquired in a data-dependent acquisition mode. Full-scan mass spectra were acquired in the Orbitrap (scan range of 350–1,500 $m/z$, resolution of 70,000, AGC target of $3 \times 10^6$, maximum injection time of 50 ms). After the MS scans, the ten most intense peaks were selected for HCD fragmentation at 30% of the normalized collision energy. HCD spectra were also acquired in the Orbitrap (resolution of 17,500, AGC target of $5 \times 10^4$, maximum injection time of 120 ms) with the first fixed mass at 180 $m/z$. Charge exclusion was selected for 1+ and 2+ ions. The dynamic exclusion was set to 5 s. All peptides were manually validated. Peptide identification and data analysis were carried out using the MASCOT Daemon client program and server (version 2.7.0)[36], and the Maxquant software (version 1.6.3.4)[37].

## Lipidomics and *N*-ret-PE identification
*N*-ret-PEs and lipids were extracted by 90% isopropanol with 0.1% formic acid from the disc membrane vesicles in dark, illuminated for 1 s and 4 s, respectively. The supernatant was transferred into a new glass tube and dried using a SpeedVac vacuum concentrator (Thermo Fisher Scientific). The evaporated lipid mixture was dissolved in a 50 μl buffer and sonicated for 10 min. For liquid chromatography–MS/MS analysis, lipids were loaded onto a C8 column (Acclaim PepMap 100, C8, inner diameter of 75 μm, particle size of 3 μm, length of 150 mm; Thermo Scientific) using a Dionex UltiMate 3000 RSLC nano System connected to an Eclipse Tribrid Orbitrap mass spectrometer (Thermo Scientific). A binary buffer system was used with buffer A of acetonitrile:H$_2$O (60:40), 10 mM ammonium formate and 0.1% formic acid, and buffer B of isopropanol:acetonitrile (90:10), 10 mM ammonium formate and 0.1% formic acid. Lipids were separated at 40 °C with a gradient from 30% to 99% buffer B at a flow rate of 300 nl min$^{-1}$ over 30 min. The electrospray voltage was set to 2.2 kV with funnel RF level at 40 and heated capillary temperature at 320 °C. For data-dependent acquisition, full MS mass range was set to 300–2,000 with a resolution of 120,000 and AGC target of 100%. Fragment spectra were acquired in the Orbitrap with a resolution of 15,000 using HCD with stepped collision energy of 25%, 30% and 35%. Phospholipids were detected in negative-ion mode. The raw data were processed by LipiDex[38] and MZmine 2 (ref. [39]) for phospholipid identification and quantification. Identification of *N*-ret-PE was processed manually. The extracted ion chromatogram of ret-PE was integrated using Xcalibur 4.1 (Thermo Fisher Scientific) and the area under the curve (AUC) was used for quantification.

## Selection and evaluation of rho-targeting molecules
Compounds used in this study were selected via high-throughput screening of a diverse library of 50,000 small molecules. The high-throughput screening approach used cells expressing tagged rho with β-galactosidase complementary and BRET detection systems. To validate the identified hits further, rho photobleaching assays were carried out along with ex vivo electrophysiology recordings[26,27].

Hit compounds F2515-3945 (**1**), F3382-0749 (**2**), F2502-0030 (**3**), F3215-0002 (**6**), F5103-0385 (**7**), F5097-2767 (**8**) and F0834-0928 (**9**) were obtained from Life Chemicals. The racemic mixture of sulconazole (**4** and **5**) was purchased from MilliporeSigma and separated as previously described on an Agilent 1100 HPLC system (Agilent Technologies) using a cellulose tris 3,5-dimethylphenyl carbamate chiralicar column (Chiral Technologies)[26].

For MS, these compounds were dissolved individually in DMSO and diluted 1,000-fold in disc membrane vesicle preparations in 200 mM ammonium acetate buffer to give a concentration for each compound of 90 μM, and a 10:1 final molar ratio of compound to rhodopsin. The disc membranes and compound were incubated for 30 min on ice, allowing for complete diffusion before native MS measurement of rho/opsin. Disc membrane supplemented with G$_t$ and PDE6 at a molar ratio of 6.5:4:1 (rho:G$_t$:PDE6) were also incubated with 90 μM compound in 200 mM ammonium acetate for 30 min on ice before native MS

measurement of G$_t$. A control experiment in the absence of membranes was performed with G$_t$ and PDE6 incubated with 90 μM compound at 4:1:50 for 30 min on ice before native MS measurement of G$_t$. Data were analysed as described above.

## Reporting summary
Further information on research design is available in the Nature Research Reporting Summary linked to this paper.

## Data availability
The data that support the findings of this study are available as follows: native MS/proteomics data and lipidomics data have been deposited to Figshare at https://doi.org/10.25446/oxford.16901326 and https://doi.org/10.6084/m9.figshare.18319361.v1, respectively.

## Code availability
The script for calculating the opsin:rho ratio in mass spectra can be accessed at https://github.com/d-que/Lig2Apo.

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

**Acknowledgements** All MS research was funded by a Wellcome Trust Investigator Award (221795/Z/20/Z) and an ERC Advanced Grant ENABLE (695511) to C.V.R. A grant from the National Institutes of Health (R01EY030912) and unrestricted grants from Research to Prevent Blindness (RPB) to the Department of Ophthalmology at UCI are gratefully acknowledged by the K.P. laboratory. We acknowledge discussions with M. Galpin and T. El-Baba on the interpretation of kinetic data. For the purpose of Open Access, the authors have applied for a CC BY public copyright license to any Author Accepted Manuscript version arising from this submission.

**Author contributions** C.V.R. and K.P. conceived the idea through the CIFAR network and provided overall supervision for the study. S.C. performed all of the experiments described in this paper under the supervision of D.S.C. and D.W. for the SoLVe and lipidomics experiments, respectively. D.Q. provided computational support for analysis. T.G., D.S. and K.P. prepared and provided ROS disc membranes and purified rho, identified rho-targeting compounds and prepared PDE6 and transducin complexes. C.V.R., S.C. and K.P. wrote the paper with contributions from other authors.

**Competing interests** K.P. is Chief Scientific Officer of Polgenix and C.V.R. is a founding director of OMass Therapeutics. All other authors declare no competing interests.

**Additional information**
**Correspondence and requests for materials** should be addressed to Dror S. Chorev, Krzysztof Palczewski or Carol V. Robinson.

**a**

cold white LED          MS inlet capillary

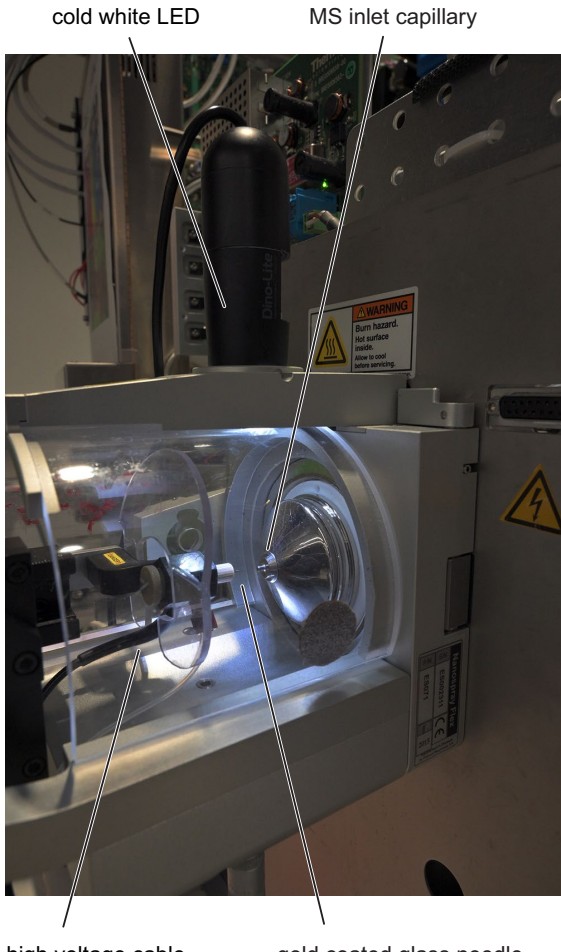

high voltage cable          gold coated glass needle

**b**

(i) needle holder  (ii) glass needle  (iii) Illumination  (iv) distance

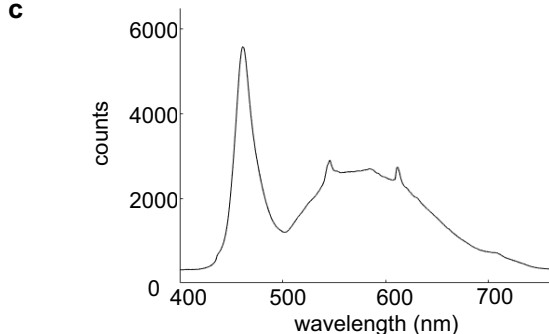

(v) high voltage cable          (vi) MS inlet capillary

**c**

Extended Data Fig. 1 | Photographic images of the electrospray set-up showing the relationship between the light source, the electrospray capillary and MS inlet and UV visible emission spectrum of the LED light source used here. a, The relationship between the electrospray interface, the cold white LED light source and the mass spectrometer. The glass capillary was displayed on the same axis of source aperture. The light stimulation set up was set orthogonally onto the capillary and the sample was at the illuminating centre. b, (i) and (ii) membrane fragments are placed in the glass needle prepared in-house and secured in the needle holder. (iii) shows the orientation of the light source orthogonal to the electrospray plume. (iv) to (vi) high voltage is applied to the needle holder and the distance between the needle tip and the MS inlet is adjusted to produce a stable electrospray ion current. Once this is achieved the electrospray interface is illuminated (t = 0) and spectra are recorded in real-time either with continuous illumination or as a function of the post illumination period. c, UV-visible emission spectrum of the cold white light source used here. The spectrum was recorded on an Andor Shamrock 303i Spectrograph equipped with an Andor Newton 920 camera. The wavelength maximum is at 460 nm.

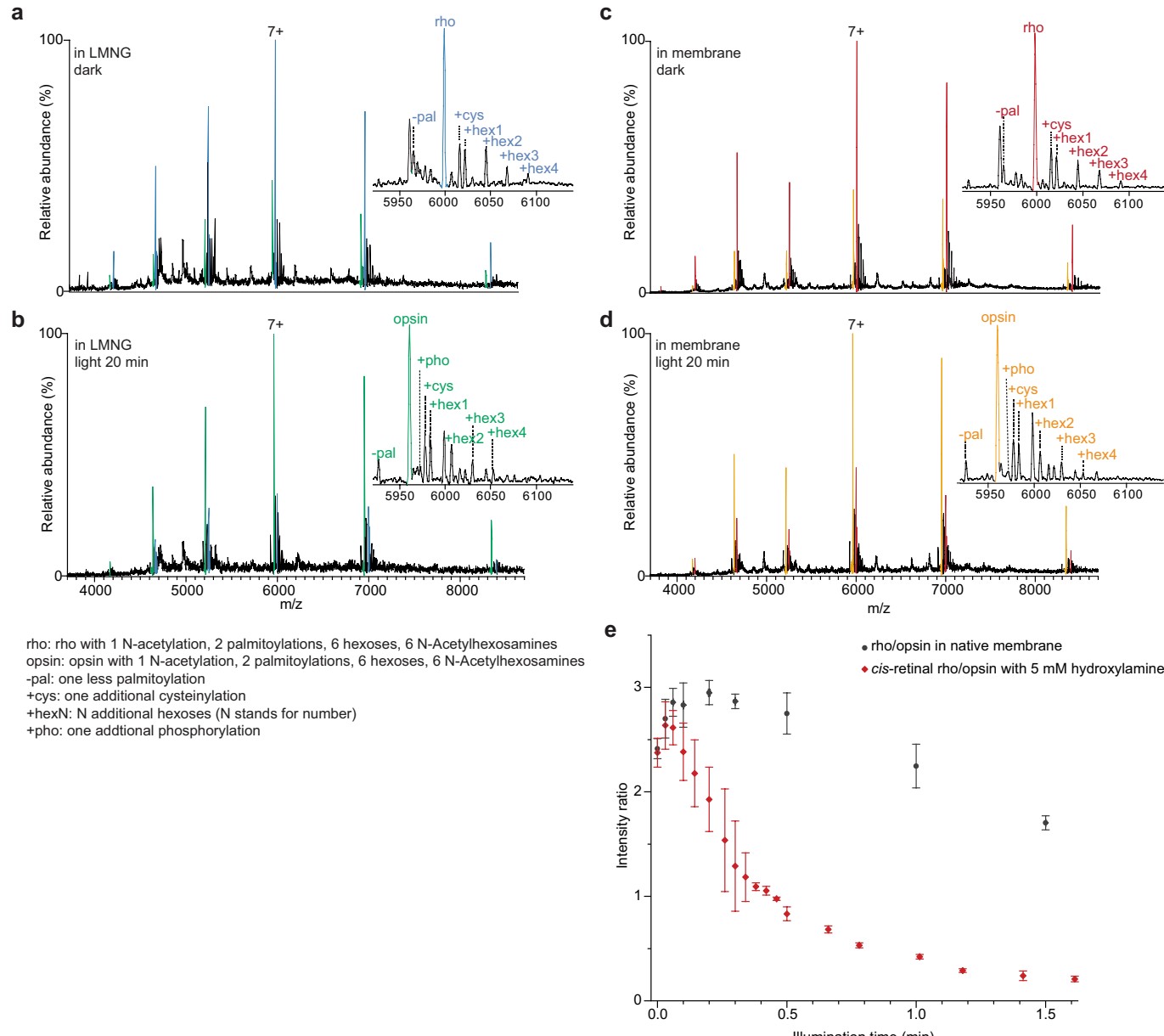

rho: rho with 1 N-acetylation, 2 palmitoylations, 6 hexoses, 6 N-Acetylhexosamines
opsin: opsin with 1 N-acetylation, 2 palmitoylations, 6 hexoses, 6 N-Acetylhexosamines
-pal: one less palmitoylation
+cys: one additional cysteinylation
+hexN: N additional hexoses (N stands for number)
+pho: one addtional phosphorylation

**Extended Data Fig. 2 | Native mass spectra of rho extracted in detergent (left-hand side) or ejected directly from membranes (right-hand side), dark-adapted or following 20 min of illumination, and decay curve following treatment with hydroxylamine. a**, From dark-adapted membranes extracted into LMNG 73% rho (blue) and 27% opsin (green) are observed in spectra initially consistent with a small population of opsin in dark-adapted bovine retinae. Inset: Expansion of the 7 + charge state reveals glycans (hexoses: hex1, hex2, hex3 and hex4), cysteinylation (cys) and palmitoylation (pal). ~95% rho /opsin has two pal sites occupied, while 5% have only one pal. **b**, After 20 min of exposure to light the population of opsin has increased and ~5% of opsin is observed with at least one phosphorylation (pho). All other

PTMs remain unchanged, 7+ charge state expanded (inset). **c**, MS of rho/opsin expelled from membrane fragments reveals 73% rho and 27% opsin, as in the detergent extraction, and an identical pattern of PTMs is observed (inset). **d**, After 20 min exposure to light in membranes the majority of rho has converted to opsin. 5% of opsin is phosphorylated while all other PTMs remain unchanged (7+ charge state (inset). **e**, Plot of the photoconversion of 11-*cis*-retinal rho to opsin in membranes either pre-treated with 5 mM hydroxylamine (red) or without pre-treatment (grey) monitored by mass spectrometry under the conditions described for Fig. 2 (main text). The experiments were performed under continuous illumination conditions and the data are presented as mean values +/− SE (n = 3).

**a**

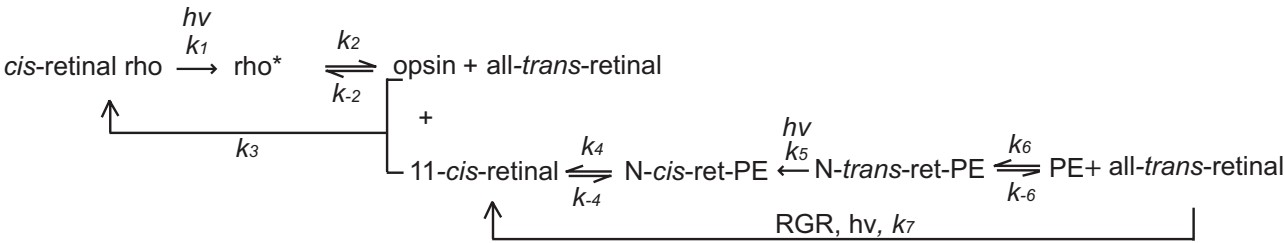

$cis$-retinal rho $\xrightarrow[k_1]{hv}$ rho* $\underset{k_{-2}}{\overset{k_2}{\rightleftharpoons}}$ opsin + all-$trans$-retinal

$+$

11-$cis$-retinal $\underset{k_{-4}}{\overset{k_4}{\rightleftharpoons}}$ N-$cis$-ret-PE $\xleftarrow[k_5]{hv}$ N-$trans$-ret-PE $\underset{k_{-6}}{\overset{k_6}{\rightleftharpoons}}$ PE+ all-$trans$-retinal

RGR, hv, $k_7$

(arrows: $k_3$ loop back to $cis$-retinal rho)

**b**

0 s - 18 s

$cis$-retinal rho $\xrightarrow[k_1]{hv}$ rho* $\xrightarrow{k_2}$ opsin + all-$trans$-retinal

$+$

11-$cis$-retinal $\underset{k_{-4}}{\overset{k_4}{\rightleftharpoons}}$ N-$cis$-ret-PE $\xleftarrow[k_5]{hv}$ N-$trans$-ret-PE

($k_3$ loop back)

**c**

$cis$-retinal rho $\xrightarrow[k_{iso}]{hv}$ rho* $\xrightarrow{k'_{hyd}}$ opsin + all-$trans$-retinal

rho $\xrightarrow{k_{hyd}}$

18 s - 180 s in LMNG or 300 s in membrane

**d**

180 s in LMNG or 300 s membrane onwards

$cis$-retinal rho $\xrightarrow[k_1]{hv}$ rho* $\underset{k_{-2}}{\overset{k_2}{\rightleftharpoons}}$ opsin + all-$trans$-retinal

$+$

11-$cis$-retinal $\underset{k_{-4}}{\overset{k_4}{\rightleftharpoons}}$ N-$cis$-ret-PE $\xleftarrow[k_5]{hv}$ N-$trans$-ret-PE $\underset{k_{-6}}{\overset{k_6}{\rightleftharpoons}}$ PE + all-$trans$-retinal

($k_3$ loop back)

**Extended Data Fig. 3 | Kinetic model for the photoconversion and regeneration of rho in disc membranes.** The photoconversion and regeneration processes involve a number of reactions outlined above. To simplify our kinetic model, we have divided the light response process into three phases. **a**, First, < 18 s when a rapid increase of rho is observed, prior to a steady decrease. **b**, This initial fast reaction 0 –18 s is attributed to the rapid regeneration of $cis$- retinal rho competing with photoconversion (rho → ops). Dark-adapted membranes possess $N$-ret-PE which responds rapidly to light such $N$-all-$trans$-ret-PE is activated and dissociates into $cis$-retinal and PE. Regenerated $cis$-retinal is then able to interact with opsin to form $cis$-retinal rho. Concurrently, existing $cis$-retinal rho photoconverts to active rho* and then deactivated opsin. **c**, During the second phase (> 18 s), after the original $N$-ret-PE is consumed, isomerization of $cis$-retinal rho to rho* continues with minimal regeneration of $cis$-retinal rho. rho* hydrolyses into opsin and all-$trans$-retinal. (Supplementary Notes).

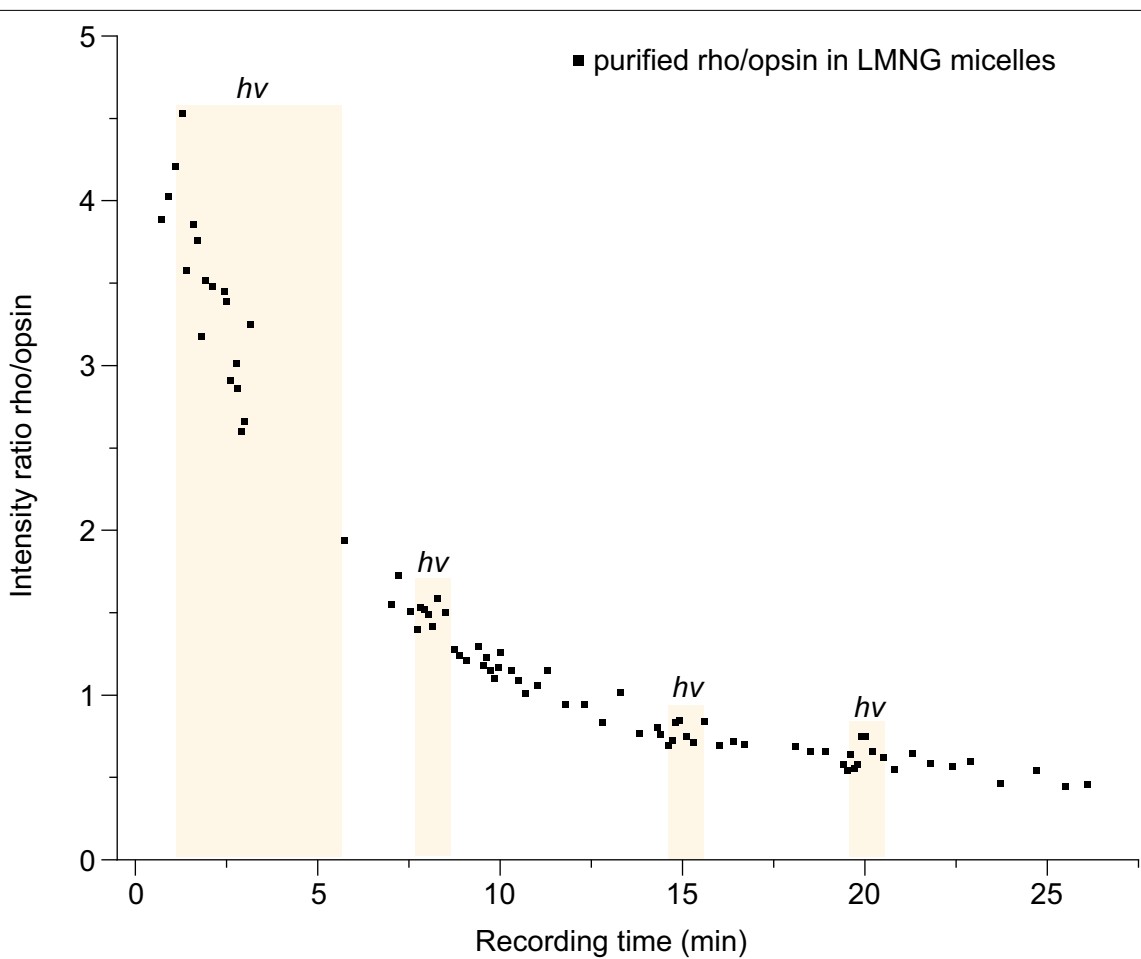

**Extended Data Fig. 4 | Monitoring the conversion of purified bovine rho to opsin through its mass change in LMNG micelles during applied illumination pulses (shaded orange) at pH 7.0 and 28 °C.** Pulses were applied at 1.1 min, 7.6 min, 15 min and 20 min for 4.6 min, 18 s, 40 s and 35 s respectively. The experimental noise makes it challenging to measure the increase in these illumination pulses in micelles compared to the experiments in the ROS membrane. However, a clear trend is observed after *circa* 20 min. The experiment was repeated twice.

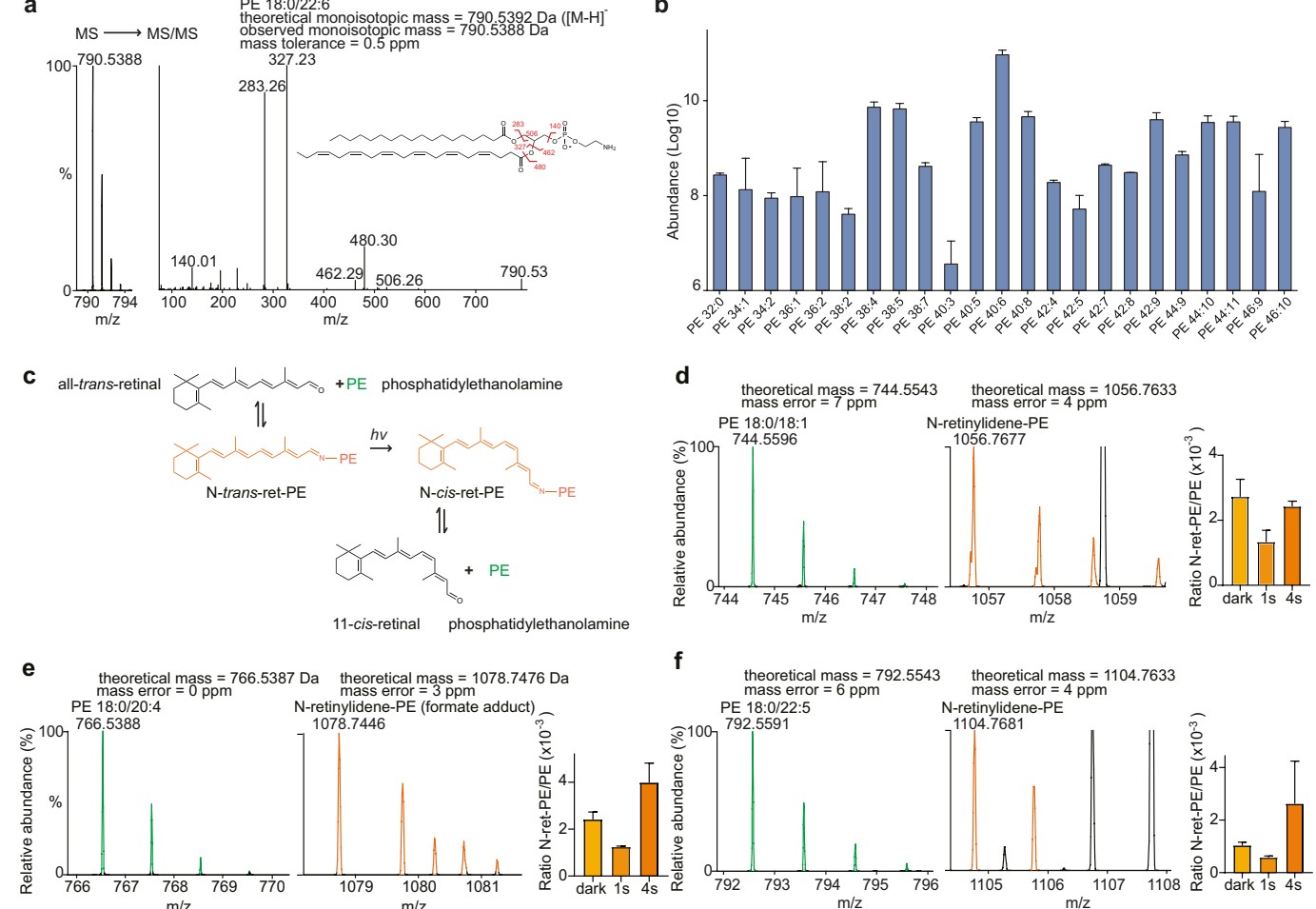

**Extended Data Fig. 5 | Extraction of lipids from the ROS membrane, identification of phosphatidylethanolamines and formation of *N*-ret-PE under controlled illumination conditions. a**, Analysis of lipids through tandem MS in the negative ion mode and assignment using the program LipiDex[38]. Shown is the fragmentation pattern of PE 40:6 confirming unsaturation in one chain. **b**, Profile and abundance of phosphatidylethanolamine in ROS membranes. Data are presented as mean values +/− SD (n = 3). A high degree of unsaturation is observed with the most abundant lipid assigned to PE 40:6. **c**, The reaction scheme of the formation of *N*-ret-PE. **d**, Detection of the formation of *N*-ret-PE 36:1. **e**, *N*-retinylidene-38:4 and **f**, *N*-retinylidene-40:5 under controlled illumination conditions and extraction into organic solvent for lipidomics. Negative ion mass spectra, monitoring the formate ion of the conjugated lipid, reveals a low intensity peak assigned through accurate mass to the *N*-retinylidene conjugate. Bar charts abundance ratio of *N*-ret-PE to free PE was monitored as a function of the illumination time and shows a clear decrease after 1 s for all three *N*-ret-PE lipids. Data are presented as mean values +/− SD (n = 3).

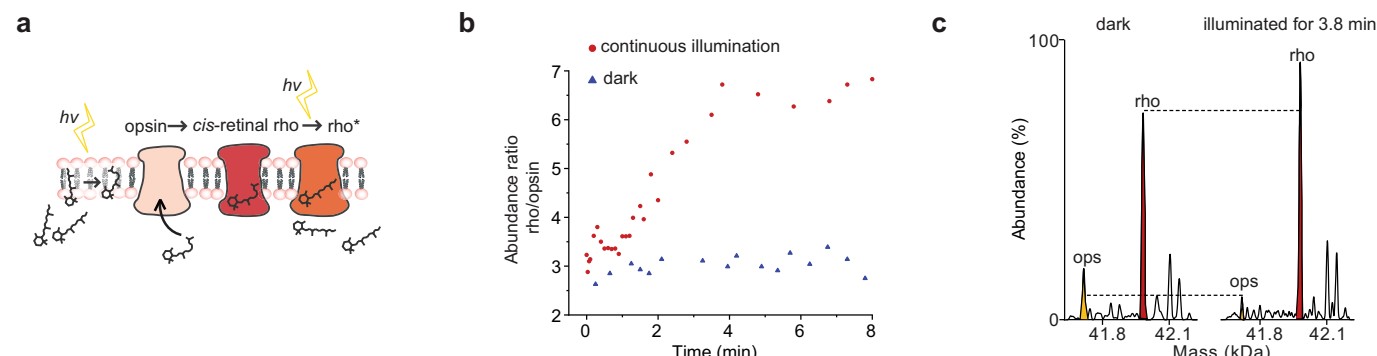

**Extended Data Fig. 6 | Rho regeneration in the presence of excess all-*trans*-retinal. a**, Schematic process of ROS opsin regenerated into rho in 10-fold excess of all-*trans*-retinal under continuous illumination. **b**, The intensity ratio of rho to opsin increases until it reaches a plateau at 3.8 min (red). Comparatively, rho remains constant in the presence of exogenous all-*trans*-retinal in the dark (blue). **c**, Zero-charge spectra of rho and opsin with 10-fold excessive all-*trans*-retinal present before and following 3.8 min illumination. This experiment was repeated twice.

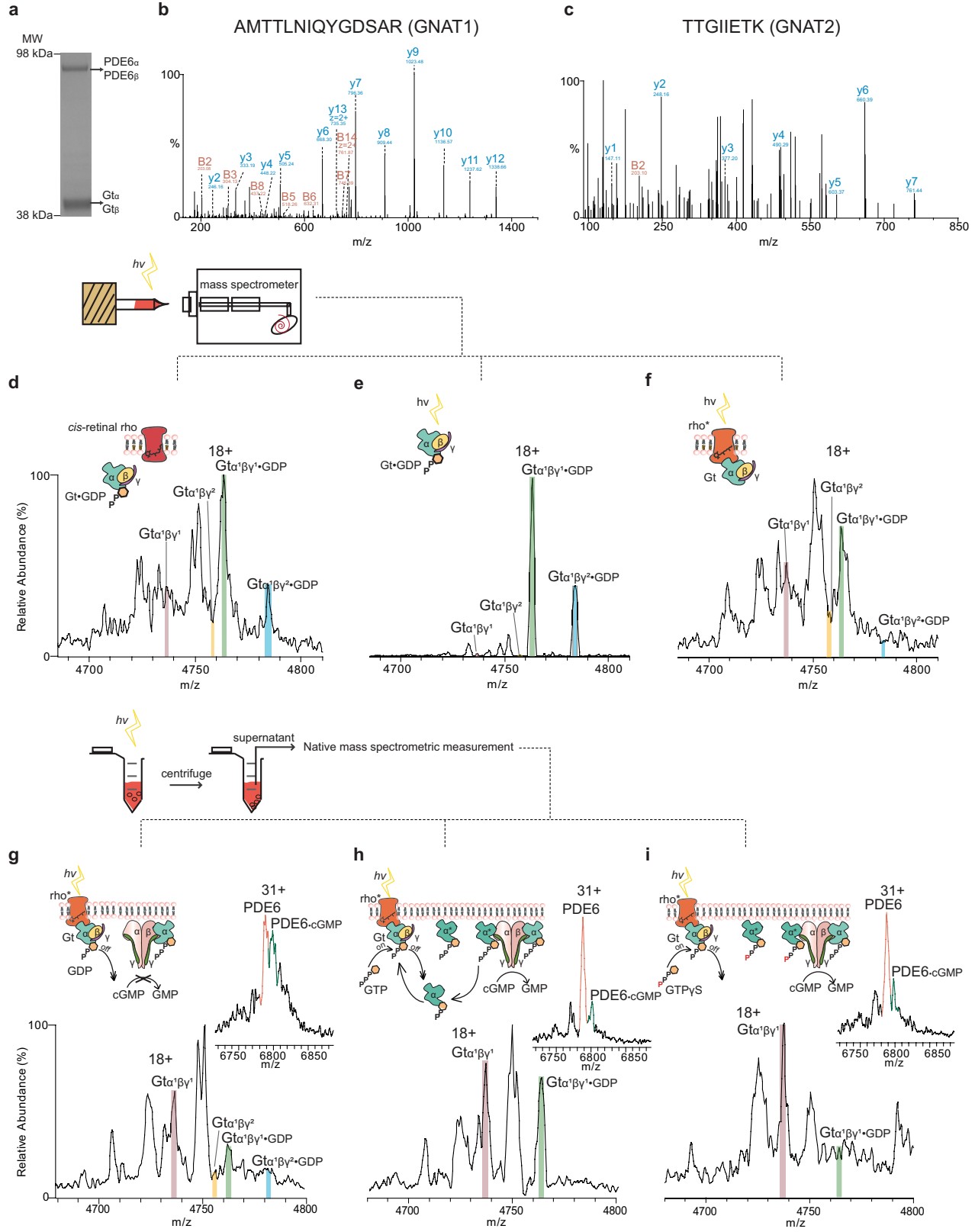

**Extended Data Fig. 7** | See next page for caption.

**Extended Data Fig. 7 | In-gel digestion proteomics and native mass spectra of $G_t$ and PDE6 complexes isolated from dark-adapted rod outer segments under different conditions. a**, Separation of the component subunits of the Gt and PDE6 complex on a 2D page gel, pre-stained with protein standards. **b**, Unique peptides of Gα1 (GNAT1) and **c**, Gα2 (GNAT2) are identified by proteomics (see Methods). **d**, Gt with rho-containing membrane in the absence of light reveals the initial extent of endogenous GDP binding to Gt (blue and green for $\gamma^1$ and $\gamma^2$ isoforms respectively) and without GDP (pink and yellow). **e**, Gt preparation illuminated in the absence of membranes was found to be bound to GDP predominantly for both isoforms. No reaction is observed in the absence of the membrane. **f**, with rho in membranes following illumination, GDP bound forms have declined in favour of *apo* states (pink and yellow). **g**, with endogenous levels of GTP, the PDE6:PDE6•cGMP complex remains ~1:1 after activation of Gt though rho*. The endogenous quantity of GTP is insufficient to produce sufficient Gα.$_{GTP}$ to interact with PDE6 and promote hydrolysis of cGMP. **h**, in the presence of an additional molar amount of GTP, $G_t$•GDP is replenished and further cGMP hydrolysis takes place. **i**, in the presence of an additional molar aliquot of GTPγs further cGMP hydrolysis has occurred but $G_t$•GDP is not replenished. The SDS-page electrophoresis (**a**) was performed twice; in-gel proteomics experiment (**b**, **c**) was performed once and the real-time native MS experiments (**d**–**f**) and bulk reactions (**g**–**i**) were repeated three times and twice respectively.

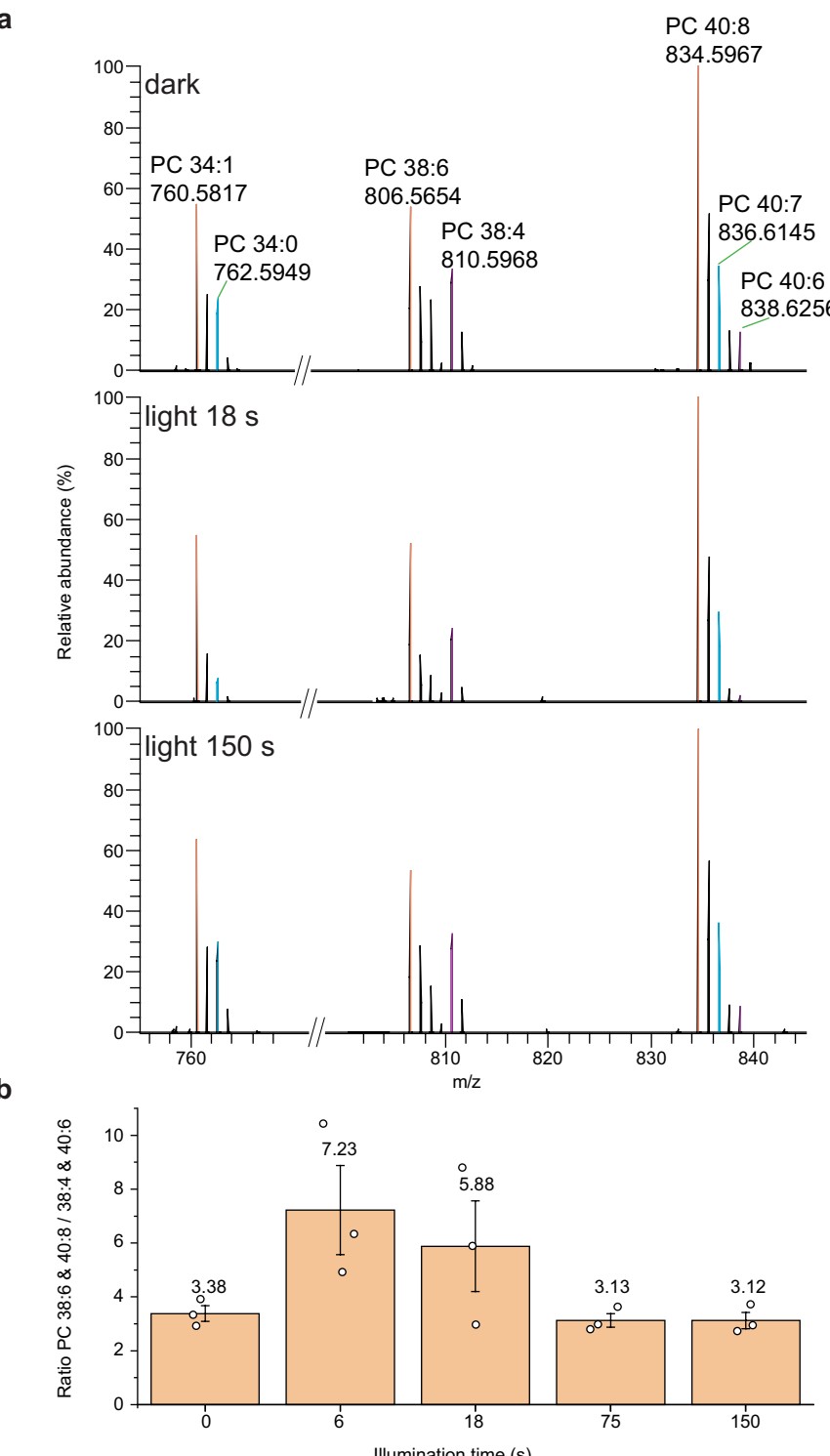

**Extended Data Fig. 8 | Monitoring the lipids that are released with rho while signalling through G$_t$ over the illumination period from dark to light. a**, PC lipids released as rho and opsin are ejected from disc membranes during signalling through Gt. Unsaturated lipids are designated as PC 34:1, 38:6 and 40:8 (orange) their more saturated counterparts 34:0, 38:4 and 40:6 (blue/purple). A decrease in the saturated lipids (blue and purple) relative to unsaturated is observed at the early stages of illumination (< 18 s). Lipids were detected in the positive ion mode. **b**, The intensities of PC 38:6 and 40:8 were manually selected to be compared with PC 38:4 and PC 40:6. Plotted is the ratio of unsaturated over saturated lipids as a function of illumination time. After 6 s and 18 s of illumination the extent of unsaturation increases. Data are presented as mean values +/− SE (n = 3).

**a**

compound **1**

compound **2**

compound **3**

compound **4**

compound **5**

compound **6**

compound **7**

compound **8**

compound **9**

**b**

**Extended Data Fig. 9 | Chemical structures of rho targeting compounds and their effects $k_{hyd}$ of rho. a**, Chemical structures of the compounds **1**–**9**. **b**, comparison of the rate of hydrolysis of rho for **1**–**9** and a control comprising ROS vesicles in 200 mM ammonium acetate with 0.1% DMSO at pH 7.0. **1**–**9** were solubilised in DMSO and diluted to give a final concentration of 90 μM in ROS vesicles in 0.1% DMSO, 200 mM ammonium acetate, pH 7.0. Data are presented as mean values +/− SE (n = 3).

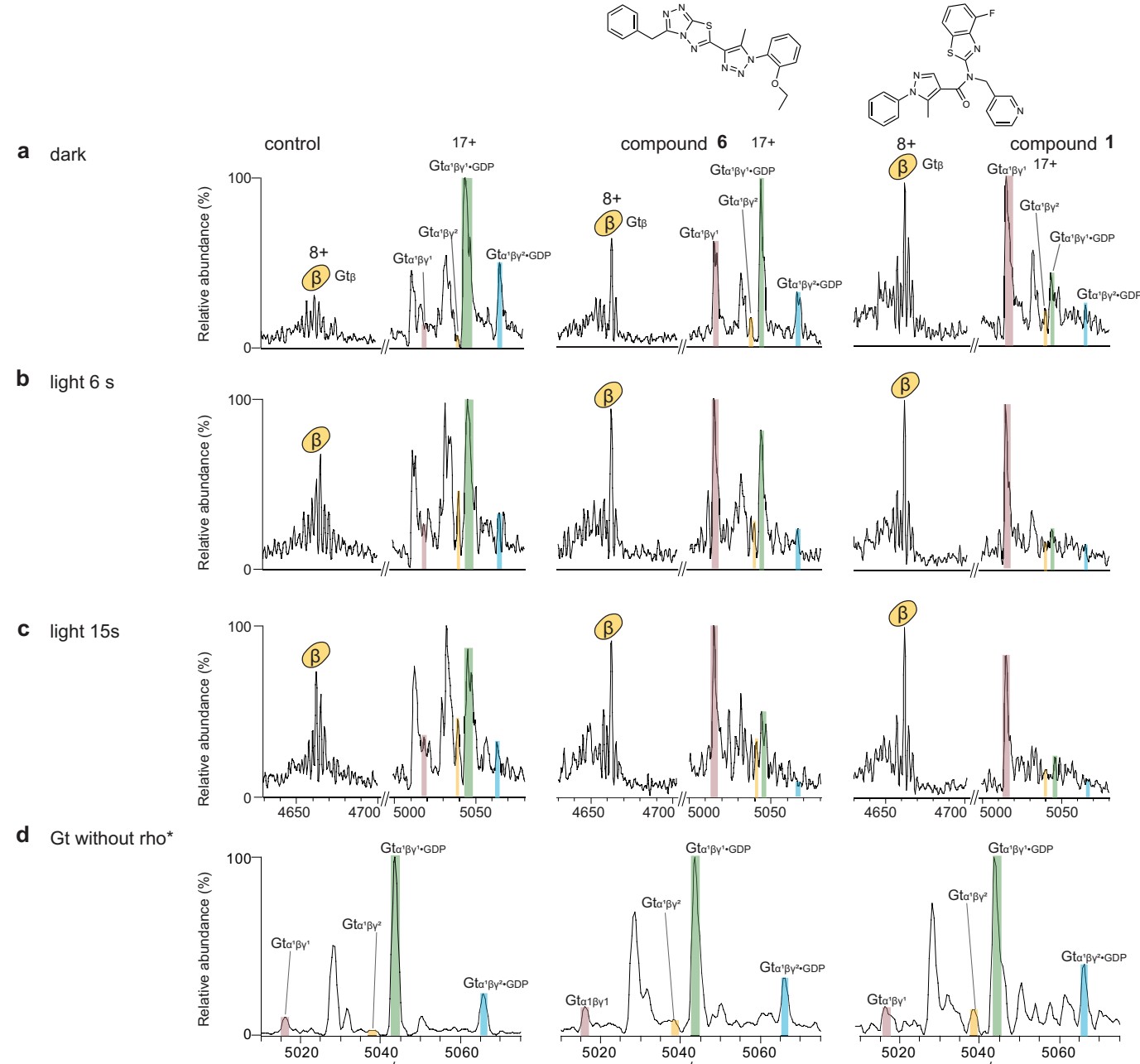

**Extended Data Fig. 10 | G$_t$ activation across native membranes during illumination of a control and in the presence of 1 and 6.** The ratio of GDP bound Gt to intermediate Gt represents a measure of the activation of transducin. Upon light absorption, rho activated G protein releases GDP **a**, at 6 s **b**, and after 15 s of the reaction. **c**, In the presence of **1** or **6** G protein signalling was accelerated and commenced even under dim red-light conditions when compared to the control experiment. Two different control

experiments were performed in the absence of compounds (left hand column) but with the same concentration of DMSO used to dissolve **1** and **6**. **d**, The second control in the presence of the same concentration of DMSO, and **1** and **6**, but in the absence of rho* establish that the agonists do not compete with the nucleotide (GDP) and that signalling occurs through rho* in the native membrane. Two replicate experiments were performed for **a**–**c** and three replicate experiments were performed for **d**.

**Extended Data Table 1 | Measured mass with standard error and abundance of ROS rho/opsin ejected from LMNG micelles and disc membranes from either dark-adapted membranes (dark) or following 20 min of illumination (light)**

| | isomers | calculated mass (Da) | LMNG mass (Da) | standard error (Da) | abundance | standard error | membrane mass (Da) | standard error (Da) | abundance | standard error |
|---|---|---|---|---|---|---|---|---|---|---|
| dark | rhodopsin,1N-acetyl, 1pal,6hex,6hexNAc | 41744 | 41743.3 | 0.1 | 5.1% | 0.2% | 41742.6 | 0.3 | 3.0% | 1.2% |
| | rhodopsin,1N-acetyl,2pal,6hex,6hexNAc | 41982 | 41981.6 | 0.1 | 35.8% | 0.9% | 41981.1 | 0.1 | 36.8% | 0.4% |
| | rhodopsin,1N-acetyl,2pal,6hex,6hexNAc,1cys | 42103 | 42103.0 | 0.2 | 10.0% | 0.2% | 42103.3 | 0.3 | 10.4% | 0.3% |
| | rhodopsin,1N-acetyl,2pal,7hex,6hexNAc | 42144 | 42144.0 | 0.2 | 8.2% | 0.3% | 42144.1 | 0.2 | 7.8% | 0.2% |
| | rhodopsin,1N-acetyl,2pal,8hex,6hexNAc | 42306 | 42306.2 | 0.1 | 8.2% | 0.6% | 42306.2 | 0.2 | 6.9% | 0.1% |
| | rhodopsin,1N-acetyl,2pal,9hex,6hexNAc | 42468 | 42468.5 | 0.2 | 4.7% | 0.3% | 42468.1 | 0.2 | 4.8% | 0.4% |
| | rhodopsin,1-Nacetyl,2pal,10hex,6hexNAc | 42630 | 42629.8 | 0.4 | 1.5% | 0.1% | 42629.5 | 0.2 | 1.4% | 0.2% |
| | opsin,1N-acetyl,1pal,6hex,6hexNAc | 41478 | 41477.3 | 0.0 | 1.0% | 0.5% | 41475.9 | 0.4 | 1.3% | 0.4% |
| | opsin,1N-acetyl,2pal,6hex,6hexNAc | 41716 | 41715.6 | 0.1 | 13.1% | 0.5% | 41715.2 | 0.0 | 14.8% | 0.7% |
| | opsin,1N-acetyl,2pal,6hex,6hexNAc,1pho | 41796 | 41796.8 | 0.5 | 1.3% | 0.1% | 41797.6 | 0.4 | 1.2% | 0.2% |
| | opsin,1N-acetyl,2pal,6hex,6hexNAc,1cys | 41837 | 41837.0 | 0.4 | 3.7% | 0.1% | 41836.4 | 0.4 | 3.6% | 0.2% |
| | opsin,1N-acetyl,2pal,7hex,6hexNAc | 41878 | 41877.9 | 0.7 | 2.4% | 0.2% | 41878.2 | 0.1 | 3.2% | 0.4% |
| | opsin,1N-acetyl,2pal,8hex,6hexNAc | 42040 | 42039.4 | 0.2 | 2.8% | 0.4% | 42040.4 | 0.3 | 2.9% | 0.6% |
| | opsin,1N-acetyl,2pal,9hex,6hexNAc | 42202 | 42202.6 | 0.3 | 1.4% | 0.1% | 42202.0 | 0.6 | 1.2% | 0.1% |
| | opsin,1N-acetyl,2pal,10hex,6hexNAc | 42364 | 42364.0 | 1.0 | 0.8% | 0.0% | 42362.1 | 0.9 | 0.7% | 0.3% |
| light | rhodopsin,1N-acetyl, 1pal,6hex,6hexNAc | 41744 | 41743.7 | 0.1 | 2.6% | 0.6% | 41741.3 | 0.7 | 2.3% | 0.5% |
| | rhodopsin,1N-acetyl,2pal,6hex,6hexNAc | 41982 | 41981.7 | 0.3 | 12.1% | 1.5% | 41981.2 | 0.2 | 14.5% | 0.6% |
| | rhodopsin,1N-acetyl,2pal,6hex,6hexNAc,1cys | 42103 | 42103.7 | 1.3 | 3.1% | 0.4% | 42104.7 | 0.3 | 4.5% | 0.6% |
| | rhodopsin,1N-acetyl,2pal,7hex,6hexNAc | 42144 | 42143.8 | 0.7 | 3.1% | 0.2% | 42143.8 | 0.1 | 3.8% | 0.6% |
| | rhodopsin,1N-acetyl,2pal,8hex,6hexNAc | 42306 | 42307.0 | 1.1 | 2.9% | 0.6% | 42306.0 | 1.2 | 2.8% | 0.3% |
| | rhodopsin,1N-acetyl,2pal,9hex,6hexNAc | 42468 | 42464.9 | 1.7 | 1.6% | 0.3% | 42468.4 | 0.2 | 2.3% | 0.4% |
| | rhodopsin,1N-acetyl,2pal,10hex,6hexNAc | 42630 | 42635.7 | 3.6 | 1.1% | 0.3% | 42632.3 | 1.5 | 1.6% | 0.3% |
| | opsin,1N-acetyl,1pal,6hex,6hexNAc | 41478 | 41477.2 | 0.7 | 5.3% | 0.2% | 41446.9 | 0.6 | 1.2% | 0.3% |
| | opsin,1N-acetyl,2pal,6hex,6hexNAc | 41716 | 41714.9 | 0.3 | 32.8% | 0.7% | 41714.8 | 0.2 | 34.9% | 2.2% |
| | opsin,1N-acetyl,2pal,6hex,6hexNAc,1pho | 41796 | 41798.6 | 0.1 | 2.2% | 0.3% | 41795.7 | 0.4 | 2.2% | 0.3% |
| | opsin,1N-acetyl,2pal,6hex,6hexNAc,1cys | 41837 | 41836.4 | 0.1 | 10.2% | 0.1% | 41836.1 | 0.1 | 9.4% | 0.2% |
| | opsin,1N-acetyl,2pal,7hex,6hexNAc | 41878 | 41878.4 | 0.4 | 8.4% | 0.4% | 41878.0 | 0.4 | 7.2% | 0.5% |
| | opsin,1N-acetyl,2pal,8hex,6hexNAc | 42040 | 42039.3 | 0.1 | 7.6% | 1.0% | 42039.6 | 0.7 | 6.9% | 0.7% |
| | opsin,1N-acetyl,2pal,9hex,6hexNAc | 42202 | 42201.3 | 0.7 | 4.7% | 0.1% | 42200.5 | 0.4 | 4.2% | 0.4% |
| | opsin,1N-acetyl,2pal,10hex,6hexNAc | 42364 | 42364.6 | 1.3 | 2.2% | 0.3% | 42364.2 | 1.1 | 2.2% | 0.3% |

The experiment was performed under continuous illumination conditions and the standard error represents the standard error of the mean across three replicate experiments.

# Reporting Summary

## Statistics

For all statistical analyses, confirm that the following items are present in the figure legend, table legend, main text, or Methods section.

| n/a | Confirmed | |
|---|---|---|
| ☐ | ☒ | The exact sample size (*n*) for each experimental group/condition, given as a discrete number and unit of measurement |
| ☒ | ☐ | A statement on whether measurements were taken from distinct samples or whether the same sample was measured repeatedly |
| ☒ | ☐ | The statistical test(s) used AND whether they are one- or two-sided<br>*Only common tests should be described solely by name; describe more complex techniques in the Methods section.* |
| ☒ | ☐ | A description of all covariates tested |
| ☐ | ☒ | A description of any assumptions or corrections, such as tests of normality and adjustment for multiple comparisons |
| ☐ | ☒ | A full description of the statistical parameters including central tendency (e.g. means) or other basic estimates (e.g. regression coefficient) AND variation (e.g. standard deviation) or associated estimates of uncertainty (e.g. confidence intervals) |
| ☒ | ☐ | For null hypothesis testing, the test statistic (e.g. *F*, *t*, *r*) with confidence intervals, effect sizes, degrees of freedom and *P* value noted<br>*Give P values as exact values whenever suitable.* |
| ☒ | ☐ | For Bayesian analysis, information on the choice of priors and Markov chain Monte Carlo settings |
| ☒ | ☐ | For hierarchical and complex designs, identification of the appropriate level for tests and full reporting of outcomes |
| ☒ | ☐ | Estimates of effect sizes (e.g. Cohen's *d*, Pearson's *r*), indicating how they were calculated |

*Our web collection on statistics for biologists contains articles on many of the points above.*

## Software and code

Policy information about availability of computer code

| | |
|---|---|
| Data collection | Thermo Xcalibur 4.1.31.9 |
| Data analysis | Thermo Xcalibur 4.1.31.9; Visual Studio Code Version 1.6.1 (Python 3.7.3 based); Code is at https://github.com/d-que/Lig2Apo; Unidec Version 2.7.3; MaxQuant Version 1.6.3.4; ImageJ bundled with 64 bit Java 1.8.0_172; OriginPro 2020 SR1 9.7.0.188; Lipidex 1.1; Mascot Verision 2.7.0; |

For manuscripts utilizing custom algorithms or software that are central to the research but not yet described in published literature, software must be made available to editors and reviewers. We strongly encourage code deposition in a community repository (e.g. GitHub). See the Nature Portfolio guidelines for submitting code & software for further information.

## Data

Policy information about availability of data

All manuscripts must include a data availability statement. This statement should provide the following information, where applicable:
- Accession codes, unique identifiers, or web links for publicly available datasets
- A description of any restrictions on data availability
- For clinical datasets or third party data, please ensure that the statement adheres to our policy

Proeomics, native mass spectrometry data and lipidomics are available through figshare: DOI: 10.25446/oxford.16901326 and (https://doi.org/10.6084/m9.figshare.18319361.v1) respectively.

# Field-specific reporting

Please select the one below that is the best fit for your research. If you are not sure, read the appropriate sections before making your selection.

☒ Life sciences  ☐ Behavioural & social sciences  ☐ Ecological, evolutionary & environmental sciences

For a reference copy of the document with all sections, see nature.com/documents/nr-reporting-summary-flat.pdf

# Life sciences study design

All studies must disclose on these points even when the disclosure is negative.

| | |
|---|---|
| Sample size | Sample size is reported |
| Data exclusions | No data has been excluded |
| Replication | Numbers of replicates is described |
| Randomization | Not relevant for this study |
| Blinding | Not relevant for this study |

# Reporting for specific materials, systems and methods

We require information from authors about some types of materials, experimental systems and methods used in many studies. Here, indicate whether each material, system or method listed is relevant to your study. If you are not sure if a list item applies to your research, read the appropriate section before selecting a response.

## Materials & experimental systems

| n/a | Involved in the study |
|---|---|
| ☒ ☐ | Antibodies |
| ☒ ☐ | Eukaryotic cell lines |
| ☒ ☐ | Palaeontology and archaeology |
| ☒ ☐ | Animals and other organisms |
| ☒ ☐ | Human research participants |
| ☒ ☐ | Clinical data |
| ☒ ☐ | Dual use research of concern |

## Methods

| n/a | Involved in the study |
|---|---|
| ☒ ☐ | ChIP-seq |
| ☒ ☐ | Flow cytometry |
| ☒ ☐ | MRI-based neuroimaging |

