## [Peer Review File · Nature]

Manuscript Title:

Capturing a rhodopsin receptor signalling cascade across a native membrane

Reviewer Comments & Author Rebuttals

Reviewer Reports on the Initial Version:

Referees' comments:

Referee #1 (Remarks to the Author):

Chen et al. analyzed rhodopsin activation cycle in real-time using mass spectrometry. They measured the ratio between rhodopsin and opsin to monitor the receptor activation rate and could calculate the isomerization rate and hydrolysis rate. The authors also analyzed GDP release and Gt association with PDE6. In the early time point, they observed rapid regeneration of rhodopsin from opsin, which is possible by involvement of the lipid bilayer.

The successful determination of receptor-G protein complex structures by CryoEM has greatly advanced our understanding of the GPCR signaling process. However, the sequential events during GPCR activation have not been studied extensively. The authors adopted native mass spectrometry to answer this important question. Real-time monitoring of membrane proteins using native mass spectrometry is very challenging, and yet the authors have successfully measured rhodopsin mass changes in real-time. However, there are a few points that should be considered before publication.

1. The authors attached a supplementary video to show the experimental set up of real-time native mass spectrometry. The authors also briefly described it in the method section and in the Video S1 caption. However, it is not easy to see due to the dim light. I understand that the authors wanted to show how light was introduced in the video, but due to the dim light, it was not easy to understand the experimental set up regarding the capillary and source aperture. It would be helpful for the readers to present pictures taken under bright light so that the readers can clearly see the experimental set up. Also, more detailed description in the method would help, too.
2. Please describe what happens to the membrane lipids or detergents while performing native mass spectrometry. Would they dissociate from the receptor?
3. In Extended Data Fig. 1, please define -pal, rho, +cys, +hex1, +hex2, +hex3, +hex4, and +pho. The readers can guess what those are, but it would be better to define them.
4. In page 4 paragraph 1, please explain how the chromophore hydrolysis (Khyd) was determined.
5. The description of Fig 2e and 2f comes earlier than that of Fig 2d in the manuscript text. Please re-number the figures.
6. The authors measured GDP-bound Gt and apo Gt in the early time point (before 15s illumination) to monitor Gt activation because apo Gt rapidly binds to GTP and GTP-Gt is converted to GDP-Gt. Please provide the timeline of GDP release, GTP binding and GTP hydrolysis to GDP in the current experimental system to justify

the monitoring of 15 s illumination. Do the authors observe GTP-bound G α if they monitor for longer time? Also, doesn't the membrane fraction already contain endogenous Gt and PDE6?

7. Related to the comment 6, to understand the kinetics of Gt activation alone, would it be possible to perform a similar experiment in the absence of PDE6 and/or in the presence of GTP γ s?

8. In Figure 3d, please describe how long the light was illuminated and provide the Gt-PDE6 complex formation kinetics (real-time monitoring) if possible.

9. Please provide the source of the compounds tested and rationale of choosing these compounds.

10. In page 6 "Unexpectedly, 1 and 6 induce transduction in the dark, implying that their enhanced isomerization rates enable them to act as potential allosteric agonists to signal through the initial trans-retinal rho population prior to illumination", the authors claims that rapid activation of Gt is due to rapid isomerization of retinal. However, the rapid activation of Gt under the dark by these compounds also could be due to direct conformational changes of rhodopsin at the cytosolic side apart from rapid isomerization. Please provide more precise mechanism of the effects of compounds 1 and 6 by measuring isomerized/hydrolyzed population and active state conformation population.

11. The experimental set up is excellent, but it is specific for rhodopsin research field. Please provide broader applications for the more general interest.

Referee #2 (Remarks to the Author):

Chen et al. report a methodological study of the rod phototransduction apparatus of the disk membrane using high performance native mass spectrometry (MS). Since the visual pigment rhodopsin is featured prominently in the paper, the work is framed as a proof-of-concept for using some variation of the methods described for G protein coupled receptor (GPCR) targeted drug discovery. In the paper, bovine rod outer segment disc membranes, which contain about 50% rhodopsin, were purified and then fragmented to create membrane fragments or vesicles, which are then directly subjected to native MS. In some experiments, the membrane fragments are supplemented with transducin (Gt), the rod cell heterotrimeric G protein, or rod phosphodiesterase (PDE6). In the membrane fragments, dark state rhodopsin (rho) was observed as a discrete peak, and the photoconversion of rho eventually to opsin could be monitored by illuminating samples and then following the evolution of the spectra over time. When the system was supplemented with purified Gt and PDE, each component of the phototransduction cascade could be observed in the MS experiments more or less as expected. For example, the guanine nucleotide binding state of Gt could be monitored and the complex formed between Gt and PDE subunits that disinhibits PDE activity could be seen. The work also addresses some issues related to the phospholipid association with rho, but not in very much detail. Finally, the authors used the MS approach to evaluate some small molecules that were previously shown to interact with rho. The main achievement described in the paper is that the native MS method can resolve the mass of a ligand bound to a typical GPCR from the GPCR (of course, the ligand in rho starts out as a covalent adduct). Whether or not the method can be used for other GPCRs is certainly not clear and is not addressed by the experiments using the rhodopsin-interacting molecules.

In general, the work is technically very strong and seeing the time-resolved conversion of rho to opsin in a disc membrane in real time in a MS experiment is exciting. However, the paper is extremely tedious and at times confusing, and many unanswered questions arise. Finally, some of the MS experiments lack appropriate direct

biochemical validation in a system that is one of the most precisely characterized biochemically in all of Nature. The bottom line is that the only thing that is measured directly in the native MS experiments is the conversion between rho and opsin. That significant achievement should not be diluted by the many confusing digressions in the paper.

Issues for clarification:

1. One of the major issues in the literature about rhodopsin is the role of rho dimerization in disk membranes. There is no mention at all about rho dimers. I assume that dimers were looked for and not seen. Are they absent? Why? Is there a technical limitation? Please discuss.
2. The description of the hydroxylamine experiments should be more precise and state that hydroxylamine can attack and cleave a Schiff base to form an oxime in a pH dependent reaction depending on the pKa of the SB.
3. The illumination conditions might be creating artifacts. It is well known that white light cannot efficiently bleach rhodopsin because of photo-back reactions and photoisomerization of free (or bound) ATR to 11-cis and 9-cis which can regenerate. In fact, the paper spends significant time defining this behavior. Why not just bleach using 500 nm light or the equivalent so as to avoid illuminating any random retinal adducts with protonated Schiff bases, which absorb at about 440 nm?
4. The type of "cold" LED light source is not given. The power stated (microW) makes no sense unless a surface area is given. Finally, in the video, it seems that there is significant illumination from the computer screen.
5. The earlier Norris et al. native MS paper (ref 7) focuses on zinc binding to rho. Was zinc seen here as well?
6. In figure 1 b, the rod cell cartoon is not correct. Disks are shown as a squiggle, not closed disks.
7. In figure 2g, the retinal structure is missing a methyl group on the C-5 carbon of the ring.
8. There is an extended section of the paper that deals with the regeneration of opsin to rho that seems to be occurring in the system. Basically, when there is a pool of opsin and all-trans-retinal is added, nothing much happens. However, if the system is illuminated, then rho appears. The authors do extraction and LC MS experiments and show the presence of phosphatidylethanolamine (PE)-all-trans-retinal (ATR) Schiff base adducts, which they claim are photo-back-converted to cis, which can then hydrolyze chemically and regenerate rho. This finding is interesting, but maybe not surprising given the illumination conditions with white light. The finding is only relevant physiologically if some role for this reaction outside of the MS experiment can be shown.
9. Also, with respect to the retinal isomerization reaction, the focus is exclusively on the PE-ATR adduct converting to 11-cis ret. However, 9-cis retinal is the more likely product, and can recombine with opsin just as well. 9-cis is not mentioned in the manuscript. Other PE type lipids besides SOPE also exist and can play a role. Although this is addressed somewhat in extended Figure 5, a much more complete analysis of these reactions would be needed to make any definitive claims. Such a study would be very challenging using the native MS approach, which can't really distinguish syn-anti or structural isomers.
10. There is a significant amount of time and effort employed to describe the kinetics of what essentially is a meta II decay assay, which can be accomplished with UV-visible or fluorescence spectroscopy. In the MS experiments a very complex 6 or more parameter model is required as described in extended Figure 3. Why is this important? It certainly is confusing and detracts from the main message.
11. The section about the rhodopsin-interacting drugs is weak and contributes very little. First, in the original Getter et al. paper, the assay used to identify the small molecules was essentially a rho dimerization disruption complementation beta-gal assay. Additional extensive validation was carried out with BRET, UV assays, Gt activation assays, etc. It is not clear that the effects described with from the MS analysis is consistent with the previously published Getter et al. paper. Did the authors originally hypothesize that they would see rho dimers in the native MS experiments that would be disrupted by the drugs? The effects of the drugs, which are used at a massive concentration of 100 micromolar, is measured only indirectly in the MS experiments. The drugs are apparently not seen directly in the MS experiments. Claims that some of the drugs have agonist activity (e.g., "1 is a more powerful agonist than 6") are not supported by the data since there is really no

pharmacological experiment. The differences between 1 and 6 are very subtle and depend on complex modeling. Figure 4f should probably be validated biochemically. Figure 4b shows no statistics – is that just one experiment for each compound? Not clear. At a minimum, an additional control experiment should be carried out with the drugs in the presence of hydroxylamine.

Minor points:

1. The abstract has a few ambiguities. Disk membranes are not studied directly. Fragments of disks are studied. What is “light-activated regeneration?” Regeneration is generally a chemical, not photochemical event. This term only becomes clear much later in the paper. “Increased saturation” makes it seem as if the saturation of PC is changing during the signaling process. Rather it is just the association of the various lipids with the rho to opsin conversion that changes.
2. Many sentences need editing, for example, the last sentence of the top paragraph in p. 4, “Since hydrolysis. . .”
3. P 12. A C10/10 column is just an empty column. What matrix was used.
4. In general, the paper is very difficult to read in part because of the inclusion in the text of multiple numerical kinetic parameters and comparison, which would be better reported in tabular format.

Referee #3 (Remarks to the Author):

A. Summary of the key results

The authors used native mass spectrometry data to document a signaling cascade for an archetypal class A GPCR across its native membrane. All three key players in the cascade can be ejected simultaneously from vesicles formed from native ROS disc membranes. They captured the conversion of rho to opsin in real-time and observed regeneration of rho following exposure to light and detected changes in the level of N-ret-PE and depletion of the conjugated chromophore following illumination. They documented changes in the unsaturation of the long chain PC lipids in the vicinity of signaling rho and the existence of conjugated retinal lipids important for regeneration. The effects of rho targeting compounds included their ability to either accelerate the rho/opsin conversion or slow down the reaction. They demonstrate the ability to modulate the Gt signaling pathway through an increase in the turnover of Gt.GDP to Gt and its subsequent dissociation to the Gtβγ complex and Gtα.GTP primed for interaction with PDE6. By capturing the entire signaling process, we demonstrated the importance of the lipid micro-environment for signaling, coupling and regeneration; and highlighted a new approach to drug discovery in which different stages of the signaling cascade can be targeted across native membranes.

B. Originality and significance: if not novel, please include reference

I believe the approach to be both original and significant.

C. Data & methodology: validity of approach, quality of data, quality of presentation

I believe the validity of the approach, the quality of the data and the quality of the presentation are all excellent, except for one caveat, the experimental temperature. It is unclear from the manuscript at what temperature experiments were performed so I assume room temperature was used. In air-conditioned buildings this is typically 24 °C. Bovine body temperature is 38.5 °C so there is a 14.5 °C difference which is substantial. Behavior of proteins on their own can be very different and when one considers the membrane/protein interaction as well this becomes a highly significant problem.

Please check the delta mass of retinal covalent attachment: I believe it is 284 minus 18.

D. Appropriate use of statistics and treatment of uncertainties

The number of samples analyzed is clearly stated and adequate.

E. Conclusions: robustness, validity, reliability

If temperature was controlled at 38.5 °C then I believe the conclusions are robust, valid and reliable. If the data was obtained at room temperature, then its value is considerably diminished.

F. Suggested improvements: experiments, data for possible revision

Experiments should be performed at the native temperature of bovine eye, 38.5 °C.

G. References: appropriate credit to previous work?

Citation of previous work on mass spectrometry of rhodopsin is suggested. MALDI is in Schey KL et al 1992;

first electrospray ionization MS was in Whitelegge JP et al 1998 and recently Marty MT et al 2021 explores the effect of Zn on rhodopsin/lipid interactions by native MS.

H. Clarity and context: lucidity of abstract/summary, appropriateness of abstract, introduction and conclusions

Clarity and context are excellent.

Author Rebuttals to Initial Comments:

Dear Reviewers,

We are grateful for the excellent review of our study and believe that the changes we have made in response to your comments have greatly improved our manuscript.

Our responses to reviewers and subsequent edits to the manuscript.

Referee #1 (Remarks to the Author):

Chen et al. analyzed rhodopsin activation cycle in real-time using mass spectrometry. They measured the ratio between rhodopsin and opsin to monitor the receptor activation rate and could calculate the isomerization rate and hydrolysis rate. The authors also analyzed GDP release and Gt association with PDE6. In the early time point, they observed rapid regeneration of rhodopsin from opsin, which is possible by involvement of the lipid bilayer.

The successful determination of receptor-G protein complex structures by CryoEM has greatly advanced our understanding of the GPCR signalling process. However, the sequential events during GPCR activation have not been studied extensively. The authors adopted native mass spectrometry to answer this important question. Real-time monitoring of membrane proteins using native mass spectrometry is very challenging, and yet the authors have successfully measured rhodopsin mass changes in real-time. However, there are a few points that should be considered before publication.

1. The authors attached a supplementary video to show the experimental set up of real-time native mass spectrometry. The authors also briefly described it in the method section and in the Video S1 caption. However, it is not easy to see due to the dim light. I understand that the authors wanted to show how light was introduced in the video, but due to the dim light, it was not easy to understand the experimental set up regarding the capillary and source aperture. It would be helpful for the readers to present pictures taken under bright light so that the readers can clearly see the experimental set up. Also, more detailed description in the method would help, too.

Thank you for this suggestion. We have elevated the brightness in the video, added further high-resolution images in bright light to show more clearly the experimental set-up (new Extended data Fig. 1 and legend) and included a detailed description of the experimental set-up and method.

2. Please describe what happens to the membrane lipids or detergents while performing native mass spectrometry. Would they dissociate from the receptor?

Under the conditions of our experiments membrane lipids and detergents are ionised along with proteins. During transmission, lipids or detergent surrounding the receptor are gradually released from the receptor via the increasing electric field and collision energy voltages. The peripheral lipids, or detergent molecules, are released first, followed by lipids that interact closely with the receptor as the activation energy within the Orbitrap is increased. We have clarified this in the text as follows:

Page 3: We identified optimal MS conditions that enable us to release rho from membrane lipids and detergents (Methods).

Methods: The following parameters were used typically and adjusted to release the membrane protein from the detergents and membrane lipids.

Legend to Figure 1c: The spectrum shown was recorded using the parameters stated above, which

led to dissociation of lipids from proteins.

3. In Extended Data Fig. 1, please define -pal, rho, +cys, +hex1, +hex2, +hex3, +hex4, and +pho. The readers can guess what those are, but it would be better to define them.

Thank you for this suggestion. We have included the full definition of these abbreviations in the legend to Extended data Fig. now 2.

4. In page 4 paragraph 1, please explain how the chromophore hydrolysis (k_{hyd}) was determined.

k_{hyd} on page 4 paragraph 1 was obtained from fitting equation 7 for the real-time decay of rho to opsin by monitoring the mass change, following illumination in real-time.

$[\rho] = [\text{cis-retinal } \rho]_0 \exp(-k_{hyd} t)$ (eq7) and the kinetic model c in extended data Fig 3.

Where t is the illumination time. $[\rho]$ and $[\text{ops}]$ are the measured abundances of rho and opsin under normal experimental conditions. $[\text{cis-retinal } \rho]_0$ is the abundance of cis-retinal rho from the dark-adapted membrane and regenerated from opsin before this phase.

We realise that confusion may have arisen since k_{hyd} was also used for the experiment following hydroxylamine treatment. We have therefore designated this measurement of hydrolysis without isomerisation as k'_{hyd} .

Page 4: We next deduced the rates of chromophore hydrolysis (k_{hyd}) in LMNG and from ROS membrane vesicles, both at pH 7.0 and 28 °C, according to the rate equation (Extended Data Fig. 3 eq. 7 and Fig. 2c).

5. The description of Fig 2e and 2f comes earlier than that of Fig 2d in the manuscript text. Please re-number the figures.

Thank you for pointing this out – figure 2 now changed to fit with the text.

6. The authors measured GDP-bound Gt and apo Gt in the early time point (before 15 s illumination) to monitor Gt activation because apo Gt rapidly binds to GTP and GTP-Gt is converted to GDP-Gt. Please provide the timeline of GDP release, GTP binding and GTP hydrolysis to GDP in the current experimental system to justify the monitoring of 15 s illumination. Do the authors observe GTP-bound Gt α if they monitor for longer time? Also, doesn't the membrane fraction already contain endogenous Gt and PDE6?

7. Related to the comment 6, to understand the kinetics of Gt activation alone, would it be possible to perform a similar experiment in the absence of PDE6 and/or in the presence of GTP γ s?

All are excellent points. We consider 6 & 7 together here:

We now include a longer timescale for the release of GDP from Gt up to 2 min with error bars from replicate experiments (Fig. 3c main manuscript).

ROS membranes were washed with isotonic and hypotonic washes to remove Gt and PDE6. Gel electrophoresis of hypotonic washes contain PDE6 and Gt (Supplementary Fig. 1). The soluble fraction containing both Gt and PDE6 was added back to the ROS membranes. The level of endogenous GTP/GDP present was sufficient to effect dissociation of Gt but only limited hydrolysis of PDE6.cGMP (Extended Data Fig. 7g).

Due to technical limitations, which necessitate removal of the membrane fragments prior to recording well-resolved mass spectra of PDE6, it was not possible to obtain a time course for this reaction. Spectra of PDE6 were recorded 30 min after exposure to light to allow for removal of membrane fragments.

We have therefore elected to perform the experiments by controlling the concentration of GTP γ S in our new extended data figure 7. Addition of one equivalent of GTP, or addition of one equivalent of GTP γ S results in further hydrolysis of PDE6.cGMP and population of Gt.GDP with GTP, but not GTP γ S. We have included this new experimental data both in the manuscript and Extended Data Fig. 7 as follows:

Page 5: To explore the relationship between rho* Gt signalling and PDE6.cGMP hydrolysis we investigated three different GTP conditions (i) with endogenous levels of GTP in the soluble fraction (ii) following addition of a molar equivalent of GTP and (iii) in the presence of a molar equivalent of non-hydrolysable GTP γ S. After illumination Gt_{GDP} levels decreased in all three cases but were replenished in (ii) the supplementary GTP experiment (Extended Data Fig. 7g-i). Linking Gt signalling with PDE6.cGMP hydrolysis, we monitored release of the hydrolysed GMP product via the ratio PDE6:PDE6.cGMP. For dark-adapted membranes, prior to illumination, an PDE6:PDE6.cGMP ratio of ~1:1 was observed consistent with full occupancy of one substrate binding site in PDE6 (Fig. 3d upper). Signalling of rho* through Gt prompts further hydrolysis of cGMP with supplementary GTP (PDE6:PDE6.cGMP ratio of ~1:0.25) compared to endogenous levels (PDE6:PDE6.cGMP ratio of ~1:0.8.5) (Extended Data Fig. 7 g, h). Since Gt _{α .GTP γ S} is also able to interact with PDE6 [1] an intermediate level of hydrolysis of cGMP was observed in the presence of an equimolar aliquot Gt _{α .GTP γ S} (PDE6:PDE6.cGMP of ~1:0.4) (Fig. 3d lower and Extended Data Fig. 7i). An additional PDE6.Gt _{α .GTP} complex with 1:1 stoichiometry can also be discerned at low intensity in the presence of Gt _{α .GTP γ S} (Fig. 3d inset). Under these experimental conditions 1:2 complexes of PDE6.(Gt _{α .GTP})₂ and PDE6.(cGMP)₂ were not observed, consistent with existing mechanistic models.

8. In Figure 3d, please describe how long the light was illuminated and provide the Gt-PDE6 complex formation kinetics (real-time monitoring) if possible.

In Fig. 3d, the sample was illuminated in the presence of the membrane for 30 min at 28 °C, then ultracentrifuged to remove membrane vesicles. This reduces the complexity and enables better resolution of the PDE6 complexes. Gt _{α .GTP}-PDE6 can be observed in the soluble fraction (purified Gt and PDE6 in the absence of membranes) implying that the Gt _{α .GTP}-PDE6 complex is a stable intermediate during cGMP hydrolysis. Currently we are unable to monitor a complete time course for the formation of the Gt _{α .GTP}-PDE6 due to the challenges of obtaining mass spectra in the presence of vesicles which necessitate the filtration step. We do however now provide an in-depth study of the effect of nucleotide concentration on page 5.

9. Please provide the source of the compounds tested and rationale of choosing these compounds. *This information has been included in the revised Methods section (pages 14 -15).*

Compounds used in this study were selected via high-throughput screening (HTS) of a diverse library of 50,000 small molecules. The HTS approach used cells expressing tagged rho with β -galactosidase complementary and BRET detection systems. To validate the identified hits further, rho photobleaching assays were carried out along with ex-vivo electrophysiology recordings^{26,27}.

Hit compounds F2515-3945 (#1), F3382-0749 (#2), F2502-0030 (#3), F3215-0002 (#6), F5103-0385 (#7), F5097-2767 (#8), F0834-0928 (#9) were obtained from Life Chemicals (ON LOS 1J0, Canada).

The racemic mixture of sulconazole (**#4 and #5**) was purchased from MilliporeSigma (Burlington, MA, USA) and separated as described previously on an Agilent 1100 HPLC system (Agilent Technologies, Santa Clara, CA, USA) using a cellulose tris 3,5 dimethylphenyl carbamate chiral column (Chiral Technologies, West Chester, PA, USA) ²⁶.

10. In page 6 “Unexpectedly, **1** and **6** induce transduction in the dark, implying that their enhanced isomerization rates enable them to act as potential allosteric agonists to signal through the initial trans-retinal rho population prior to illumination”, the authors claim that rapid activation of Gt is due to rapid isomerization of retinal. However, the rapid activation of Gt under the dark by these compounds also could be due to direct conformational changes of rhodopsin at the cytosolic side apart from rapid isomerization. Please provide more precise mechanism of the effects of compounds **1** and **6** by measuring isomerized/hydrolyzed population and active state conformation population.

*This is a good point. However, UV absorbance of rho in the presence of compounds **1** and **6**, either in the dark or illuminated state, showed no spectral shifts that could be attributed to binding of the molecules at the cytosolic side [3]. Moreover, the intrinsic fluorescence of rho-Trp²⁶⁵, assigned to the chromophore release mechanism, was significantly reduced following illumination implying a reduction in the reaction rate. Together these results suggest that these molecules are acting as allosteric modulators, with no perturbation of the chromophore-binding site, but with changes in the hydrolysis rate.*

*We also present the isomerised/hydrolysed populations in Fig. 4d (main manuscript) and compare this data with the control under the same conditions. We find that while **1** has a marginal effect on the rate of hydrolysis, **6** decelerates this step. To measure active states, we used engagement with Gt (Extended Data Fig. 10). In the control experiment Gt_{GDP} in the dark persists with little dissociation to apo Gt. By contrast in the presence of **1** or **6** apo GT predominates after 6 s of light exposure with **1** and **6**. We conclude therefore that while both increase signalling via transducin, **6** retains retinal and continues signalling through Gt following isomerisation.*

Page 6: After 15 min, noticeable differences occur in the rho/opsin conversion which is faster when modulated by **1** than **6**, with **1** being marginally faster than the control. Since we see no evidence for displacement of retinal by these compounds, or changes in the conformation of rhodopsin [3], we speculate that **1** and **6** act as allosteric modulators. Calculating rates of hydrolysis and isomerization in the presence of hydroxylamine we find that **1** accelerates both hydrolysis and isomerization while **6** has an accelerated isomerization rate but retains retinal by decelerating hydrolysis, thereby potentially maintaining active signalling states for extended periods.

11. The experimental set up is excellent, but it is specific for rhodopsin research field. Please provide broader applications for the more general interest.

We have focused on rod phototransduction and rhodopsin because of the experimental advantages and the fact that it can be studied directly from native membrane environments without chemical extraction. To address the reviewer’s suggestion, we have added the following text:

Page 7: We anticipate that similar approaches will be applicable to other cell surface receptors (e.g. olfactory receptors) present at high density enabling kinetic studies of molecular events, and identification of endogenous and synthetic receptor ligands that perturb signalling.

Referee #2 (Remarks to the Author):

Chen et al. report a methodological study of the rod phototransduction apparatus of the disk membrane using high performance native mass spectrometry (MS). Since the visual pigment rhodopsin is featured prominently in the paper, the work is framed as a proof-of-concept for using some variation of the methods described for G protein coupled receptor (GPCR) targeted drug discovery. In the paper, bovine rod outer segment disc membranes, which contain about 50% rhodopsin, were purified and then fragmented to create membrane fragments or vesicles, which are then directly subjected to native MS. In some experiments, the membrane fragments are supplemented with transducin (Gt), the rod cell heterotrimeric G protein, or rod phosphodiesterase (PDE6). In the membrane fragments, dark state rhodopsin (ρ) was observed as a discrete peak, and the photoconversion of ρ eventually to opsin could be monitored by illuminating samples and then following the evolution of the spectra over time.

When the system was supplemented with purified Gt and PDE, each component of the phototransduction cascade could be observed in the MS experiments more or less as expected. For example, the guanine nucleotide binding state of Gt could be monitored and the complex formed between Gt and PDE subunits that disinhibits PDE activity could be seen. The work also addresses some issues related to the phospholipid association with ρ , but not in very much detail. Finally, the authors used the MS approach to evaluate some small molecules that were previously shown to interact with ρ . The main achievement described in the paper is that the native MS method can resolve the mass of a ligand bound to a typical GPCR from the GPCR (of course, the ligand in ρ starts out as a covalent adduct). Whether or not the method can be used for other GPCRs is certainly not clear and is not addressed by the experiments using the rhodopsin-interacting molecules. In general, the work is technically very strong and seeing the time-resolved conversion of ρ to opsin in a disc membrane in real time in a MS experiment is exciting. However, the paper is extremely tedious and at times confusing, and many unanswered questions arise. Finally, some of the MS experiments lack appropriate direct biochemical validation in a system that is one of the most precisely characterized biochemically in all of Nature. The bottom line is that the only thing that is measured directly in the native MS experiments is the conversion between ρ and opsin. That significant achievement should not be diluted by the many confusing digressions in the paper.

Issues for clarification:

1. One of the major issues in the literature about rhodopsin is the role of ρ dimerization in disk membranes. There is no mention at all about ρ dimers. I assume that dimers were looked for and not seen. Are they absent? Why? Is there a technical limitation? Please discuss.

This is an important point and we acknowledge previous excellent discussion about the role of rhodopsin dimerization in signalling. However, we did not observe dimerization in our experimental set up and these were not reported in other MS papers, including the research of Norris et al. Different crystal and cryo-EM structures of rhodopsin in nanodiscs reveal a relatively small and "fluid" dimer interface, maintained mainly by hydrophobic residues, consistent with a weak and reversible dimerization in the native ROS disc membrane (Zhao et al, JBC 2019 Sep 27;294(39):14215-14230). It is possible that rhodopsin dimers are formed transiently but not observed in our approach. We have included discussion of this important point in our revised manuscript.

Page 3: Under these conditions monomeric ρ was ejected. No dimeric population was observed, attributed to the relatively small dimer interface, observed *via* cryo-EM in nanodiscs [4], that dissociates under these MS conditions.

2. The description of the hydroxylamine experiments should be more precise and state that hydroxylamine can attack and cleave a Schiff base to form an oxime in a pH dependent reaction

depending on the pKa of the SB.

Thank you for this suggestion we have now clarified this discussion as follows:

Page 4: Rhodopsin in its ground-state is insensitive to hydroxylamine but, upon photoisomerisation of *cis*-retinal to all-*trans*-retinal, hydroxylamine is able to attack and cleave the Schiff base in the photoactivated rhodopsin to form all-*trans*-retinyl oxime. [5]

3. The illumination conditions might be creating artifacts. It is well known that white light cannot efficiently bleach rhodopsin because of photo-back reactions and photoisomerization of free (or bound) ATR to 11-*cis* and 9-*cis* which can regenerate. In fact, the paper spends significant time defining this behavior. Why not just bleach using 500 nm light or the equivalent so as to avoid illuminating any random retinal adducts with protonated Schiff bases, which absorb at about 440 nm?

This is an important point. Primarily we are interested in investigating rhodopsin photoactivation and regeneration in a near-native membrane environment, under natural white light. We have therefore used a sophisticated kinetic model to study photoactivation and hydrolysis in white light. Photo-transformation of rhodopsin in green light exhibits similar trends (Fig. 1, for reviewer #2).

*Slightly faster photo-transformation is observed under green light than in white light due to the absence of the *cis*-retinal generation from retinal-lipid adducts under green light conditions. We performed the analogous experiment (Fig. 2f main text) monitoring rho-opsin conversion following pulses of green light at discrete time points. Except for the first pulse, we did not observe increased population of rho at every pulse of green light (c.f. Fig. 2f for white light comparison). We attribute the rise of rho during the first pulse to low populations of RGR. The lack of increased rho at subsequent light pulses is attributed to the absence of the *cis*-retinal generation from retinal-lipid adducts under green light.*

Finally we believe the richness of our data lies in the fact that we can look at the photo-back reactions within the context of the membrane environment. We wanted to address the possibility of retinal adducts that could be involved in regeneration and believe that this is an interesting aspect of our study in native membranes under natural white light.

Figure 1 Investigation of photoconversion under green light conditions. **a**, The same experimental set-up as described in the main paper was used to measure the conversion of rho to opsin in native membrane fragments under either white or green light conditions. The decay is marginally faster under green light conditions due to the absence of cis-retinal generation from retinal-lipid adducts. **b**, Monitoring the rho-opsin photo conversion following pulses of green light at discrete time points.

4. The type of “cold” LED light source is not given. The power stated (microW) makes no sense unless a surface area is given. Finally, in the video, it seems that there is significant illumination from the computer screen.

The surface area of illumination is 3 mm² and encompasses the nanoflow needle tip where the photoconversion takes place. We have included this information in the new experimental set-up together with detailed photographs of the experimental set-up as well as a UV-vis spectrum of the light source used (new Extended Data Fig. 1). We agree that there is significant illumination from the computer screen however, this does not initiate the photoconversion reaction. No change in the rho population under these conditions is observed prior to illumination with the LED (Extended data Fig. 6b, (blue)).

5. The earlier Norris et al. native MS paper (ref 7 now ref 14) focuses on zinc binding to rho. Was zinc seen here as well?

We often observe GPCRs bound to Zn following detergent extraction. The Norris paper focuses on detergent extracted rho. If we add exogenous Zn we are able to detect Zn binding. However, in this case from the native membrane fragments, we did not see Zn binding.

6. In figure 1 b, the rod cell cartoon is not correct. Disks are shown as a squiggle, not closed disks. *We apologise for this mistake – thank you for pointing this out. Now fixed.*

7. In figure 2g, the retinal structure is missing a methyl group on the C-5 carbon of the ring. *Now corrected - with apologies!*

8. There is an extended section of the paper that deals with the regeneration of opsin to rho that seems to be occurring in the system. Basically, when there is a pool of opsin and all-trans-retinal is added, nothing much happens. However, if the system is illuminated, then rho appears. The authors do extraction and LC MS experiments and show the presence of phosphatidylethanolamine (PE)-all-trans-retinal (ATR) Schiff base adducts, which they claim are photo-back-converted to cis, which can then hydrolyze chemically and regenerate rho. This finding is interesting, but maybe not surprising given the illumination conditions with white light. The finding is only relevant physiologically if some role for this reaction outside of the MS experiment can be shown.

It was believed for some time that rhodopsin regeneration involves the visual cycle (RPE65 retinoid isomerase). It was also proposed that, at least in part, regeneration involves the RGR protein [6-8]. However, most recent studies suggest that there must be a third mechanism by which the chromophore can be produced [9]. The most logical explanation could be through the retinyl-phospholipid intermediates as proposed by Kaylor and further validated in our study [10].

In detail the Travis' lab has shown that blue light (450-nm) also converts all-trans-retinal specifically to 11-cis-retinal through a retinyl-PE intermediate in photoreceptor membranes. The quantum efficiency of the photoconversion of N-ret-PE Schiff base adducts is similar to that of rho. However, monochromatic light is not required for retinyl lipid photoregeneration/ photoisomerization. In fact, white light is far more effective at photoisomerization of N-ret-PE than narrow-band 450-nm light. Given their overlapping spectra, photoisomerization of N-ret-PE and bleaching of rhodopsin occur

simultaneously in natural light [10]. That same study [10] showed the physiological relevance in that live mice regenerate rhodopsin more rapidly in blue light while retinas and isolated cone cells show increased photosensitivity following exposure to blue light. These results indicate that light contributes to visual-pigment renewal in mammalian rods and cones through a non-enzymatic process involving retinyl-phospholipids. Our data provides further molecular detail this mechanism.

Page 5: Together these results imply that conversion of opsin to rho in membranes can be supplemented by addition and photo-activation of all-*trans*-retinal and that PE conjugation provides a source of retinal for regeneration in membranes, a mechanism that has been suggested for sustained vision in daylight [10].

9. Also, with respect to the retinal isomerization reaction, the focus is exclusively on the PE-ATR adduct converting to 11-*cis* ret. However, 9-*cis* retinal is the more likely product, and can recombine with opsin just as well. 9-*cis* is not mentioned in the manuscript. Other PE type lipids besides SOPE also exist and can play a role. Although this is addressed somewhat in extended Figure 5, a much more complete analysis of these reactions would be needed to make any definitive claims. Such a study would be very challenging using the native MS approach, which can't really distinguish syn-anti or structural isomers.

*We accept that a limitation of our MS approach is that it cannot distinguish different retinal isomers. However, in the same article mentioned above [10] the preferred cis-isomer formed after illumination of all-*trans*-retinal adducts was found to be 11-*cis*-retinal (85-86%), followed by 9-*cis*-retinal in much lower abundances (12-13%) and 13-*cis*-retinal (1-2%).*

We now include the possibility that other ATR-PE adducts might also be present at low levels (page 4):

Previous reports have implicated an N-all-*trans*-retinyl-PE conjugate that can undergo photoisomerisation to form *cis*-retinals (primarily 11 *cis*-retinal (85-86%) followed by 9-*cis*-retinal (12-13%) and 13- *cis*-retinal (1-2%)[10]. Our MS approach cannot distinguish these structural isomers but our lipidomics data does reveal high concentrations of polyunsaturated fatty acids (PUFA) including PE (Extended Data Fig. 5a-c).

We have also broadened our analysis of PE conjugates including a further example (revised Extended Data Fig 5 d-f). We are however restricted by the considerable overlap within the mass spectrum since most ret-PE conjugates are present in the range 1000-1100 Da. This range is close to the masses of NAPE (N-acylphosphatidylethanolamine).

10. There is a significant amount of time and effort employed to describe the kinetics of what essentially is a meta II decay assay, which can be accomplished with UV-visible or fluorescence spectroscopy. In the MS experiments a very complex 6 or more parameter model is required as described in extended Figure 3. Why is this important? It certainly is confusing and detracts from the main message.

We accept that there are many descriptions of kinetic models that have been published that are simpler than this one. However, all previous models describe results from detergent micelles or membrane mimetics. This is the first measurement from within the membrane with the complexity of regeneration mechanisms. This is the primary reason for the additional steps in the mechanism. Moreover, we are interested in measuring rhodopsin photoactivation in natural/white light. It is important to have a sophisticated kinetic model consider to as many events as possible. This is fundamental to further investigate the photoisomerisation and hydrolysis, in a single measurement, with or without hydroxylamine.

11. The section about the rhodopsin-interacting drugs is weak and contributes very little. First, in the original Getter et al. paper, the assay used to identify the small molecules was essentially a rho dimerization disruption complementation beta-gal assay. Additional extensive validation was carried out with BRET, UV assays, Gt activation assays, etc. It is not clear that the effects described with from the MS analysis is consistent with the previously published Getter et al. paper. Did the authors originally hypothesize that they would see rho dimers in the native MS experiments that would be disrupted by the drugs? The effects of the drugs, which are used at a massive concentration of 100 micromolar, is measured only indirectly in the MS experiments. The drugs are apparently not seen directly in the MS experiments. Claims that some of the drugs have agonist activity (e.g., “1 is a more powerful agonist than 6”) are not supported by the data since there is really no pharmacological experiment. The differences between 1 and 6 are very subtle and depend on complex modelling.

We agree the difference between these molecules is small. It is also important to point out the ligands are not yet drugs but were selected based on assays designed to measure disruption of rho dimerization [2, 3]. We believe this part of the study is important because it highlights what is possible in terms of providing starting points for further development of ligands. We have modified the text to make this point more clearly.

We studied their effects in two separate assays (i) rho->opsin photoconversion and (ii) signalling through transducin via release of GDP. While we would not expect concordance between different assays the ligands identified previously as affecting dimerization did show reproducible effects in our assay, particularly for some dimerisation ‘disrupters’ which also showed decelerated rates of hydrolysis.

For the photoconversion assay error bars have been included for the 3 replicate experiments. Moreover, rather than requiring complex modelling - effects can be seen directly in the raw data (Fig. 4b,c).

After 3 min with 6 rho remains predominate while for 1, which is close to the control measurement, opsin predominates. For the second assay however 1 is very much faster in releasing GDP than the control or 6 (Extended Data Fig. 10).

*We have clarified in the text that these are **ligands not drugs** identified via high throughput screening (Methods pages 14 -15) with promising characteristics for further development. We also provide more information about their selection and have removed claims of their agonist effects in the absence of pharmacology.*

Page 7: We conclude that **1** and **6** target rho and amplify signalling through Gt, making them ideal starting points for further chemical development.

Figure 4f should probably be validated biochemically. Figure 4b shows no statistics – is that just one experiment for each compound? Not clear. At a minimum, an additional control experiment should be carried out with the drugs in the presence of hydroxylamine.

As suggested we attempted a biochemical validation of Figure 4f, employing a plate reader to monitor release of fluorescent GDP. Although this assay works well for recombinant GPCRs with a mini G protein, the time scale of our experiment is (<15 s) and the requirement to control light exposure within the plate reader, make this an extremely challenging experiment practically. Moreover, disk membrane fragments are heterogeneous, likely containing other specific or non-specific GDP binding proteins. These factors, coupled with the practical challenges, led to irreproducible results.

However the additional control experiment that you suggested works well - the hydroxylamine treatment in the presence of the ligands. This experiment allows us to separate the effects of hydrolysis and isomerisation and is reproducible across three replicate experiments, error bars are also included (Fig. 4d).

Page 6: Calculating rates of hydrolysis and isomerization in the presence of hydroxylamine we find that **1** accelerates both hydrolysis and isomerization while rho* bound to **6** has an accelerated isomerization rate but retains retinal by decelerating hydrolysis, thereby potentially maintaining active signalling states for extended periods.

Finally, we believe these results highlight our ability to identify ligands in two different assays (photoconversion and phototransduction) with different effects, and serve as examples of the different assays that are now possible in the context of fragments of disk membranes.

Minor points:

1. The abstract has a few ambiguities. Disk membranes are not studied directly. Fragments of disks are studied. What is "light-activated regeneration?" Regeneration is generally a chemical, not photochemical event. This term only becomes clear much later in the paper. "Increased saturation" makes it seem as if the saturation of PC is changing during the signaling process. Rather it is just the association of the various lipids with the rho to opsin conversion that changes.

We agree with all these points and thank the reviewer for their careful reading. All are now corrected and fragments of disk membranes is used throughout.

Abstract: Considering the lipids ejected with rhodopsin, we demonstrate that opsin can be regenerated in membranes through photoisomerised retinal-lipid conjugates and provide evidence for an increase in the unsaturation of the long-chain phosphatidylcholine bound to rhodopsin during signalling.

2. Many sentences need editing, for example, the last sentence of the top paragraph in p. 4, "Since hydrolysis. . ."

3. P 12. A C10/10 column is just an empty column. What matrix was used. *This section has been rewritten for clarity.*

4. In general, the paper is very difficult to read in part because of the inclusion in the text of multiple numerical kinetic parameters and comparison, which would be better reported in tabular format. *We agree with this suggestion. All numerical values now appear with the graphs from which they are deduced. We believe this simplifies the text and we thank the reviewer for this suggestion.*

Referee #3 (Remarks to the Author):

A. Summary of the key results

The authors used native mass spectrometry data document to characterize a signaling cascade for an archetypal class A GPCR across its native membrane. All three key players in the cascade can be ejected simultaneously from vesicles formed from native ROS disc membranes. They captured the conversion of rho to opsin in real-time and observed regeneration of rho following exposure to light and detected changes in the level of N-ret-PE and depletion of the conjugated chromophore following illumination. They documented changes in the unsaturation of the long chain PC lipids in the vicinity of signaling rho and the existence of conjugated retinal lipids important for regeneration. The effects of rho targeting compounds included their ability to either accelerate the rho/opsin conversion or slow down the reaction. They demonstrate the ability to modulate the Gt signaling pathway through an increase in the turnover of Gt.GDP to Gt and its subsequent dissociation to the Gt $\beta\gamma$ complex and

Gt α .GTP primed for interaction with PDE6. By capturing the entire signaling process, we demonstrated the importance of the lipid micro-environment for signaling, coupling and regeneration; and highlighted a new approach to drug discovery in which different stages of the signaling cascade can be targeted across native membranes.

B. Originality and significance: if not novel, please include reference

I believe the approach to be both original and significant.

Thank you for this supportive comment.

C. Data & methodology: validity of approach, quality of data, quality of presentation

I believe the validity of the approach, the quality of the data and the quality of the presentation are all excellent, except for one caveat, the experimental temperature. It is unclear from the manuscript at what temperature experiments were performed so I assume room temperature was used. In air-conditioned buildings this is typically 24 °C. Bovine body temperature is 38.5 °C so there is a 14.5 °C difference which is substantial. Behavior of proteins on their own can be very different and when one considers the membrane/protein interaction as well this becomes a highly significant problem. Please check the delta mass of retinal covalent attachment: I believe it is 284 minus 18.

The mass we have calculated should be the zero-charge mass. We agree with the reviewer that this mass difference is therefore 266. Thank you for correcting this point.

D. Appropriate use of statistics and treatment of uncertainties

The number of samples analyzed is clearly stated and adequate.

E. Conclusions: robustness, validity, reliability

If temperature was controlled at 38.5 °C then I believe the conclusions are robust, valid and reliable. If the data was obtained at room temperature, then its value is considerably diminished.

We did explore the effects of temperature on the hydrolysis reaction using a designed heated nanoflow capillary set-up (originally reported [11] and recently adapted). Practically however this is extremely challenging for membrane fragments which need to be heated in the capillary holder causing needle blockage. While a higher temperature promoted faster kinetics there was a significant loss in reproducibility between replicate experiments; presumably associated with the difficulties of ensuring that this elevated temperature was maintained throughout the time course of the reaction. We therefore opted to use a controlled reaction temperature of 28 °C which provided the most stable experimental data for this challenging system. It is also worth noting that virtually all previous rhodopsin kinetic experiments were carried out at 15 °C to 25 °C depending on the optimal temperature for the experimental set-up.

F. Suggested improvements: experiments, data for possible revision

Experiments should be performed at the native temperature of bovine eye, 38.5 °C. *We include a single time point at higher temperature to show that the experiment can be performed at the higher temperature but the mass spectrometry data is less reproducible (Figure 2, for reviewer #3).*

Figure 2. Comparison of rho/opsin ejected from membranes in the dark (upper) and following exposure to light for 21 min at 28 °C (l.h.s.) and 40 °C (r. h.s.). The difference in signal to noise level and stability are apparent between these two datasets recorded at different temperatures.

G. References: appropriate credit to previous work?

Citation of previous work on mass spectrometry of rhodopsin is suggested. MALDI is in Schey KL et al 1992; first electrospray ionization MS was in Whitelegge JP et al 1998 and recently Marty MT et al 2021 explores the effect of Zn on rhodopsin/lipid interactions by native MS.

All previous MS work now included (refs. 12-14).

H. Clarity and context: lucidity of abstract/summary, appropriateness of abstract, introduction and conclusions

Clarity and context are excellent.

Thank you!

References

1. Wang, X., Plachetzki, D.C., Cote, R.H.: The N termini of the inhibitory γ -subunits of phosphodiesterase-6 (PDE6) from rod and cone photoreceptors differentially regulate transducin-mediated PDE6 activation. *J Biol Chem.* **294**, 8351-8360 (2019)

2. Getter, T., Gulati, S., Zimmerman, R., Chen, Y., Vinberg, F., Palczewski, K.: Stereospecific modulation of dimeric rhodopsin. **FASEB J.** **33**, 9526-9539 (2019)
3. Getter, T., Kemp, A., Vinberg, F., Palczewski, K.: Small molecule modulators of the signaling state of vertebrate rhodopsin **J. Biol. Chem.** **in press**,
4. Zhao, D.Y., Pöge, M., Morizumi, T., Gulati, S., Van Eps, N., Zhang, J., Miszta, P., Filipek, S., Mahamid, J., Plitzko, J.M., Baumeister, W., Ernst, O.P., Palczewski, K.: Cryo-EM structure of the native rhodopsin dimer in nanodiscs. **J Biol Chem.** **294**, 14215-14230 (2019)
5. Sakmar, T.P., Franke, R.R., Khorana, H.G.: Glutamic acid-113 serves as the retinylidene Schiff base counterion in bovine rhodopsin. **Proc Natl Acad Sci U S A.** **86**, 8309-8313 (1989)
6. Morshedean, A., Kaylor, J.J., Ng, S.Y., Tsan, A., Frederiksen, R., Xu, T., Yuan, L., Sampath, A.P., Radu, R.A., Fain, G.L., Travis, G.H.: Light-Driven Regeneration of Cone Visual Pigments through a Mechanism Involving RGR Opsin in Müller Glial Cells. **Neuron.** **102**, 1172-1183.e1175 (2019)
7. Zhang, J., Choi, E.H., Tworak, A., Salom, D., Leinonen, H., Sander, C.L., Hoang, T.V., Handa, J.T., Blackshaw, S., Palczewska, G., Kiser, P.D., Palczewski, K.: Photocatalytic generation of 11-cis-retinal in bovine retinal pigment epithelium. **J Biol Chem.** **294**, 19137-19154 (2019)
8. Ward, R., Kaylor, J.J., Cobice, D.F., Pepe, D.A., McGarrigle, E.M., Brockerhoff, S.E., Hurley, J.B., Travis, G.H., Kennedy, B.N.: Non-photopic and photopic visual cycles differentially regulate immediate, early, and late phases of cone photoreceptor-mediated vision. **J Biol Chem.** **295**, 6482-6497 (2020)
9. Frederiksen, R., Morshedean, A., Tripathy, S.A., Xu, T., Travis, G.H., Fain, G.L., Sampath, A.P.: Rod Photoreceptors Avoid Saturation in Bright Light by the Movement of the G Protein Transducin. **J Neurosci.** **41**, 3320-3330 (2021)
10. Kaylor, J.J., Xu, T., Ingram, N.T., Tsan, A., Hakobyan, H., Fain, G.L., Travis, G.H.: Blue light regenerates functional visual pigments in mammals through a retinyl-phospholipid intermediate. **Nat Commun.** **8**, 16 (2017)
11. Benesch, J.L., Sobott, F., Robinson, C.V.: Thermal dissociation of multimeric protein complexes by using nanoelectrospray mass spectrometry. **Anal Chem.** **75**, 2208-2214. (2003)

Reviewer Reports on the First Revision:

Referees' comments:

Referee #1 (Remarks to the Author):

The authors clarified all the issues that I have raised in the previous review. One minor comment is that the quality of Extended Fig 7 needs to be improved.

Referee #2 (Remarks to the Author):

Chen et al. have submitted a revised manuscript that addresses the earlier concerns of this reviewer. In particular, experimental details have been clarified, the figures have been improved, the lamp spectrum is now provided, photobleaching conditions have been clarified, ambiguities of language have been removed, the precision of the abstract has been greatly enhanced, certain important definitions have been added, and all minor errors noted in text and figures have been corrected. The rebuttal document is one of the best I have seen in terms of tone and completeness. I have no further comments about the manuscript. The work is groundbreaking -- a really amazing technology applied to the complex, but understandable vertebrate phototransduction machinery.

Referee #3 (Remarks to the Author):

I am happy with author revisions and have no further comments.